# STRUCTURED REASONING FOR LLMS: A UNIFIED FRAMEWORK FOR EFFICIENCY AND EXPLAINABILITY

**Yubo Dong, Hehe Fan,**∗ **Linchao Zhu, Yi Yang**
College of Computer Science and Technology
Zhejiang University
Hangzhou, China

## ABSTRACT

Recent Large Language Models (LLMs) have made remarkable progress, but they still struggle with complex reasoning tasks such as logical deduction and planning. This is partly because they rely primarily on token-level probability relationships, which limits their ability to reason effectively. In this paper, inspired by cognitive science and neurosymbolic AI, we introduce **Structured Reasoning**, which aimes at enhancing the reasoning capabilities of LLMs from the step level. To this end, we first collect high-frequency, domain-agnostic reasoning step tags and construct a structured reasoning dataset with those tags. Then, we treat a reasoning process as a *directed acyclic graph*, where the vertices represent steps and the edges indicate the direction of reasoning. In this context, an efficient reasoning process corresponds to, or can be characterized by, a sparse reasoning graph. To construct reasoning graphs, we introduce *structured tags* for reliable step extraction from LLM outputs. For single-graph optimization, we propose the *MaxFlow reward*, which rewards graphs with balanced node contributions and fewer redundant steps. The quality of a sparse reasoning graph can be reflected by the total flow from all steps to the final answer. For multi-graph comparison, we propose the *LCS reward*, which selects reliable reasoning paths by identifying optimal common subsequences (consecutive steps) shared across multiple generated responses (sequences). Experiments with DeepSeek-R1-Distill-Qwen-1.5B and 7B models show that our method consistently outperforms GRPO and other carefully tuned baselines across various context lengths (0.5k–8k). Structured Reasoning shows particular strength in efficiency (better performance with fewer steps) and stability (consistently generating high-quality outputs across a temperature range of 0.1 to 1.0). Methods and examples is currently available on our website: Structured-Reasoning.

## 1 INTRODUCTION

Large Language Models (LLMs) such as DeepSeek-R1 (DeepSeek-AI et al., 2025a), OpenAI-o1 (OpenAI, 2024), and QwQ (QwQ, 2025) have rapidly advanced the state of natural language processing, knowledge access, and automated decision support. Despite their impressive language capabilities and broad applicability, the existing reasoning patterns suffer from several limitations: (i) **redundant** and verbose content, (ii) **unstable** performance, and (iii) **poor interpretability** of internal reasoning logic. These challenges hinder LLMs' safety, controllability, and trustworthiness in practical applications.

We posit that advancing trustworthy reasoning in LLMs requires a transition to explicitly structured, auditable processes. Structured reasoning, inspired by cognitive science and dual-process theories (Bronkhorst et al., 2022; Forstmann et al., 2016; Evans, 2018; Miller & Cohen, 2001), breaks down problem solving into clear steps with specific purposes (like restating the problem and checking the answer). Making these steps explicit and central to the process helps in several ways: (i) keeps the reasoning focused and on-track, (ii) allows us to evaluate each step properly, (iii) makes it easier to understand how different layers of the model handle these steps.

This paper proposes a novel approach to enhance LLMs with structured reasoning capabilities, inspired by cognitive science theories and recent advances in neurosymbolic artificial intelligence.

---

∗Corresponding author. This work was supported by the National Natural Science Foundation of China (No. 92570101).

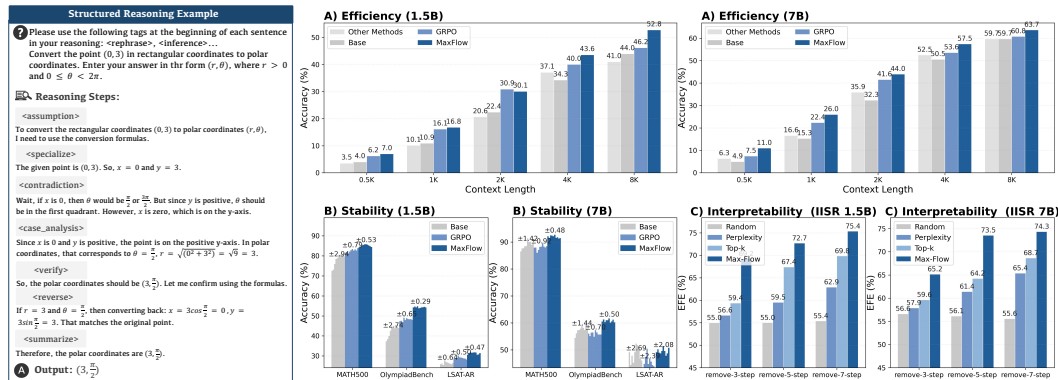

Figure 1: Structured reasoning improves efficiency, stability, and interpretability. Left: an example of our structured reasoning process. Right: across general tasks, combining structured data tuning with structure-aware optimization outperforms GRPO and other baselines in (i) efficiency (fewer, denser steps), (ii) stability (robust across temperatures), and (iii) interpretability (clear step dependencies).

Specifically, we introduce mechanisms that explicitly encode structured knowledge representations and reasoning processes in LLMs. Then, we treat reasoning processes as directed graphs, where the vertices represent steps and the edges indicate the direction of reasoning, leveraging both the flexibility of neural networks and the interpretability and precision of symbolic reasoning.

Our framework first transforms unstructured data into structured format by incorporating explicit reasoning step tags that clearly indicate each step of the reasoning process. These structured annotations enable adaptive fine-tuning that helps models develop systematic reasoning patterns. Additionally, we implement a layer-wise dependency tracing procedure using step-to-step attention matrices, enabling detailed analysis of reasoning relationships within the LLM's computation process.

To further enhance reasoning efficiency, we extend Group Relative Policy Optimization (GRPO) Shao et al. (2024b) with two structure-aware algorithms: (1) *MAX-Flow*: Constructs sparse reasoning graphs by analyzing step-to-step attention matrices and measures the quality of the graph based on each step's contribution to the final answer, (2) *Longest Common Subsequence (LCS)*: Improves reasoning quality by identifying optimal common subsequences across multiple generated responses and leveraging these consistent steps as reliable reasoning paths. Our contributions are as follows:

- We propose a novel Structured Reasoning approach that achieves more concise reasoning and stable performance, demonstrating significant improvements in efficiency (better performance at shorter lengths), stability (consistent quality across temperatures 0.1-1.0), and interpretability across various scenarios on DeepSeek-R1-Distill-Qwen models.

- We develop a method to automatically extract common reasoning patterns and convert unstructured reasoning into structured formats, creating a dataset that helps transform free-form reasoning steps into well-organized structured reasoning chains.

- We propose an attention-based layer-wise analysis framework that constructs step-to-step attention maps across model layers, providing enhanced interpretability of reasoning steps and revealing that middle layers play a crucial role in integrating broader reasoning context.

- We enhance GRPO with two complementary algorithms: 1) MAX-Flow, which constructs sparse reasoning graphs by analyzing step-to-step attention matrices and measures each step's contribution to the final answer, and 2) LCS, which improves reasoning quality by identifying optimal common subsequences across multiple generated responses and leveraging these consistent steps as reliable reasoning paths.

## 2 RELATED WORK

**Reinforcement Learning Helps Efficiency Improvement** Recent approaches use RL to improve reasoning efficiency, from basic length penalties (Team et al., 2025; Li et al., 2025; Arora & Zanette,

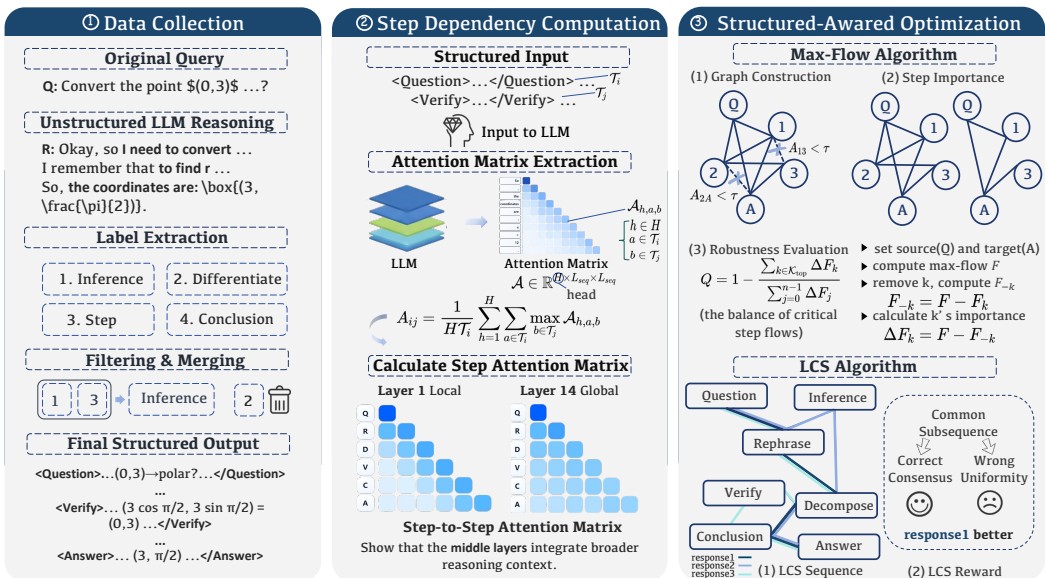

Figure 2: Illustration of our three-stage pipeline for enhancing LLMs with Structured Reasoning. (1) Data Collection: Extract structured reasoning labels from unstructured LLM responses, producing outputs with structured tags. (2) Step Dependency Computation: Compute step attention matrices to construct reasoning directed graph. (3) Structure-Aware Opitmization: Apply Max-Flow algorithm for providing a significantly more accurate understanding of reasoning step dependencies and LCS algorithm for improving reasoning quality by identifying optimal common subsequences across multiple generated responses and leveraging these consistent steps as reliable reasoning paths.

2025) to more sophisticated methods. L1 (Aggarwal & Welleck, 2025) embeds length constraints in training instructions, while O1-Pruner (Luo et al., 2025a) balances brevity and accuracy against reference benchmarks. DAST (Shen et al., 2025b) introduces adaptive reasoning through token-length budget, allocating resources based on problem complexity. THINKPRUNE (Hou et al., 2025) employs a length-aware reward with tightening constraints, while Think When You Need (Yang et al., 2025) uses comparative rewards to guide models toward concise yet effective solutions.

**Efficient CoT According to Perplexity** Several works optimize reasoning chains using perplexity-based methods (Jelinek et al., 2005), including stepwise refinement (Cui et al., 2025b), token pruning (Xia et al., 2025), attack detection (Alon & Kamfonas, 2023) and step elimination strategies (Liu et al., 2024). Furthermore, (Zhang et al., 2025) proposes exploration based on entropy for multistep reasoning. Our research reveals that perplexity metrics inadequately assess the importance of reasoning steps, demonstrating that our MAX-Flow algorithm outperforms perplexity-based approaches in evaluating the importance of reasoning steps.

**Language Model Reasoning (for Math)** Since OpenAI-O1 (Jaech et al., 2024), followed by O3 (OpenAI, 2025) and DeepSeek-R1 (DeepSeek-AI, 2025), researchers have proposed increasingly sophisticated RL algorithms, including Dr.GRPO (Liu et al., 2025b), LCPO (Aggarwal & Welleck, 2025), REINFORCE++ (Hu, 2025), DAPO (Yu et al., 2025), DPO-VP (Tu et al., 2025), VinePPO (Kazemnejad et al., 2024), CPPO (Lin et al., 2025), VAPO (Yue et al., 2025), and GRO (Cai, 2025). Empirical investigations have explored data scaling (Shen et al., 2025a), curriculum strategies (Wen et al., 2025; Roux et al., 2025), and reward engineering (Gao et al., 2024b; Cui et al., 2025a; Ma et al., 2023). Recent evaluations (Hochlehnert et al., 2025) show many reported improvements fail against properly optimized baselines. Our methods evaluate across multiple seeds to ensure reproducibility.

## 3 MOTIVATION: REASONING AS FLOW ON STRUCTURED GRAPHS

We propose to view the reasoning process as a single-source single-sink flow diffusion process from the question step to the answer step, as illustrated in Figure 2. This perspective transforms

the challenge of optimizing redundant reasoning steps and improving efficiency into a problem of optimizing reasoning graph structure.

**Single Reasoning Graph Perspective.** For reasoning steps that are redundant, repetitive, or meaningless, both the answer step and intermediate conclusion steps tend to ignore them, resulting in weaker connections for these step nodes to the final answer (sink). Conversely, consider an ideal case of a strictly step-by-step dependent CoT reasoning process: each new intermediate inference step strongly depends on the previous step. In such cases, removing any single reasoning step would interrupt the flow, making each step's contribution approximately equal. A high-quality reasoning process should thus exhibit **balanced step contributions**, where no single step dominates the flow, indicating a robust, non-redundant reasoning chain.

**Multi-Graph Comparison Perspective.** When comparing multiple reasoning graphs that reach the correct answers, we can optimize by identifying common attention edges. Under an ideal assumption, if one reasoning graph's path is a subset of another's, it appears more concise in reasoning logic. Furthermore, if two graphs have identical reasoning paths, we generally consider the process with shorter corresponding reasoning steps to be more efficient. This motivates our LCS-based reward that encourages alignment with correct completions with length suppression.

## 4 METHOD

### 4.1 STRUCTURED REASONING DATA COLLECTION

Due to the free-form nature of reasoning passages, small LLMs struggle to reliably parse them into discrete reasoning steps. To address this, we design a pipeline to construct structured reasoning data with explicit step labels.

Given a question set $\mathcal{Q}_0$ and a teacher model $T$ (DeepSeek-R1), for each $q \in \mathcal{Q}_0$ we obtain raw reasoning $r^{\text{raw}}$ and answer $a$, then elicit a linear label chain $\mathbf{l} = (l_1 \to \cdots \to l_m)$ via a self-summarization prompt A.8. We keep frequent labels, merge synonyms, and remove domain-specific ones to form the core set. Let $\mathcal{P}$ be the set of the labels.

To synthesize aligned structured traces over the questions $\mathcal{Q}$, we sample labels $\pi \in \mathcal{P}$ for each $q$ and prompt $T$ to generate a labeled reasoning $r$, producing the raw structured set $\mathcal{D}_{\text{raw}} = \{(q, \pi, r, a)\}$. We apply a filtering function $F(q, \pi, r, a)$ to verify answer correctness and reasoning difficulty. The final corpus is:

$$\mathcal{D}_{\text{struct}} = \{(q, \pi, r, a) \in \mathcal{D}_{\text{raw}} \mid F(q, \pi, r, a) = 0\}. \tag{1}$$

This produces a tiny but high-quality dataset suitable for structured tuning. We tuned models to produce structured reasoning under a designed prompt template that enforces explicit reasoning labels. For each Question-Reasoning-Answer triplet $(q, r, a)$ in the dataset $\mathcal{D}_{\text{struct}}$, the question $q$ is combined with our structured reasoning prompt $I$, which guides the model to use specific reasoning labels at the start of each sentence. The model learns to generate the structured reasoning $r$ and the answer $a$. The structured model, $\theta_{\text{struct}}$, is trained as follows:

$$\theta_{\text{struct}} = \prod_{(q, r, a \in \mathcal{D}_{\text{struct}})} P(r, a \mid q, I), \tag{2}$$

where $I$ denotes our structured prompt and $\mathcal{D}_{\text{struct}}$ is the set of selected high-quality samples.

### 4.2 LAYER-WISE STEP-DEPENDENT TRACING

**Step-to-Step Attention Matrix.** Given a layer attention tensor $\mathcal{A} \in \mathbb{R}^{H \times L_{seq} \times L_{seq}}$ ($H$ heads, sequence length $L_{seq}$), we compute the normalized step attention matrix $\mathbf{A} \in \mathbb{R}^{n \times n}$ for $n$ reasoning steps. For steps $i, j$ with token ranges $[s_i^{\text{start}}, s_i^{\text{end}}]$ and $[s_j^{\text{start}}, s_j^{\text{end}}]$ respectively:

$$A_{ij} = \frac{1}{H \mathcal{T}_i} \sum_{h=1}^{H} \sum_{a \in \mathcal{T}_i} \max_{b \in \mathcal{T}_j} \mathcal{A}_{h,a,b}, \tag{3}$$

where $\mathcal{T}_k = [s_k^{\text{start}}, s_k^{\text{end}}]$ denotes the token range of step $k$.

We denote the token range of step $k$ by $\mathcal{T}_k = [s_k^{\text{start}}, s_k^{\text{end}}]$. The time complexity of this procedure is $\mathcal{O}(B \times H \times n^2 \times T_{\text{avg}})$, where $T_{\text{avg}}$ is the average number of tokens per step. In practice, the inner max is computed with vectorized reductions, and per-layer intermediate buffers are released immediately, keeping memory footprint low.

### 4.3 Reinforcement Learning for Improved Structured Reasoning

We assume that a better reasoning process should have fewer unnecessary connections between steps.

**1. Max-Flow Reward** We assess step importance via a max-flow/min-cut based reward (Ford & Fulkerson, 1956). Let the induced reasoning graph have $V = n$ nodes and $E$ retained edges after thresholding (edge density $\rho = E/n^2 \ll 1$). We employ a sparse max-flow implementation (Push–Relabel style with standard heuristics).

**(a) Graph Construction.** Construct directed graph $G = (V, E)$. Nodes $V = \{1, ..., n\}$ representing steps (node 1: Question, node $n$: Answer); Edges $(i, j) \in E$ with capacity $A_{ij}$ when $A_{ij} > \tau$ (threshold $\tau = 0.05$). Thresholding prunes weak edges, yielding sparser, quasilinear backbones that accelerate flow computation while preserving salient reasoning channels.

**(b) Step Importance.** For source $s = 1$ and target $t = n$: Compute max-flow $F$ in $G$ using Ford-Fulkerson algorithm. For each node $k \in V \setminus \{s, t\}$, $\Delta F_k = F - F_{-k}$, where $F_{-k}$ is the max-flow in subgraph $G_{-k}$ (node $k$ removed). The value $\Delta F_k$ quantifies how crucial step $k$ is for reaching the conclusion.

**(c) Robustness Evaluation.** Let $\mathcal{K}_{\text{top}}$ be the top-25% most important steps. The reasoning quality metric $Q \in [0, 1]$ is computed as:

$$Q = 1 - \frac{\sum_{k \in \mathcal{K}_{\text{top}}} \Delta F_k}{\sum_{j=0}^{n-1} \Delta F_j}, \tag{4}$$

The resoning reward $r_{\text{maxflow}} = Q$ if correct, else $-1$. Higher $Q$ indicates more balanced reasoning.

**Time Complexity.** The theoretical worst-case complexity for max-flow is $\mathcal{O}(V^2 E)$ using Dinic's algorithm. For dense attention graphs where $E = \Theta(n^2)$, this yields $\mathcal{O}(n^4)$. However, our two-stage optimization (optimized Dinic + residual network reuse) achieves **7.41× speedup**, with empirical complexity between $\mathcal{O}(n^2 \log n)$ and $\mathcal{O}(n^{2.5})$. The overall time complexity is:

$$\mathcal{O}(BHn^2 T_{\text{avg}}) + \mathcal{O}(n^{2.5}), \tag{5}$$

with detailed analysis in Appendix F.

**2. The Longest Common Subsequence Reward** The LCS reward requires at least one correct completion as reference. Given sampled reasoning completions $\mathcal{R} = \{r_1, \ldots, r_n\}$ for a question, let $r_{\text{acc}}(r_i)$ denote the correctness reward for reasoning completion $r_i$. For each pair $(r_i, r_j)$, we extract their reasoning steps and compute the longest common subsequence (LCS) of reasoning labels, denoted $\text{LCS}(r_i, r_j)$.

Let $L_{\text{lcs}}$ be the total length of the matched steps in the LCS and $L_i$ be the total length of steps in $r_i$.

To prevent *length hacking* (i.e., artificially increasing the token count of each reasoning step to inflate scores), for each matched step $k$ in the LCS with lengths $\ell_{i,k}$ and $\ell_{j,k}$, we introduce a length suppression factor, defined as $\text{ratio}_k = \frac{\ell_{j,k}}{2\ell_{i,k}}$ if $\ell_{i,k} > \ell_{j,k}$ and $\text{ratio}_k = 1 - \frac{\ell_{i,k}}{2\ell_{j,k}}$ otherwise. Subsequently, the length of the weighted LCS is defined as $L_{\text{lcs}} = \sum_{k \in \text{LCS}(r_i, r_j)} \text{ratio}_k \cdot \ell_{i,k}$. We define the pairwise LCS score as:

$$\text{Score}_{\text{lcs}}(r_i, r_j) = \begin{cases} \frac{L_{\text{lcs}}}{L_i}, & \text{if both } r_i \text{ and } r_j \text{ are correct,} \\ -\frac{L_{\text{lcs}}}{L_i}, & \text{if both } r_i \text{ and } r_j \text{ are incorrect,} \\ 1 - \frac{L_{\text{lcs}}}{L_i}, & \text{if } r_i \text{ is correct, } r_j \text{ is incorrect,} \\ -1 + \frac{L_{\text{lcs}}}{L_i}, & \text{if } r_i \text{ is incorrect, } r_j \text{ is correct.} \end{cases} \tag{6}$$

Here, a higher LCS ratio is rewarded when compared with correct completions (encouraging consensus on high-quality reasoning), while a lower LCS ratio is rewarded when compared with incorrect

completions (encouraging diversity from incorrect reasoning). The length suppression factor $\text{ratio}_k$ penalizes unnecessarily long steps and encourages concise reasoning.

Finally, the overall LCS reasoning reward for $c_i$ is averaged over all other completions:

$$r_{\text{lcs}}(c_i) = \frac{1}{n-1} \sum_{j \neq i} \text{Score}_{\text{lcs}}(c_i, c_j). \tag{7}$$

**Time Complexity.** For sequences of lengths $L_1$ and $L_2$, the code fills a DP table and backtracks, costing $\mathcal{O}(L_1 L_2)$ time; the weight pass over LCS matches is $\mathcal{O}(\min\{L_1, L_2\})$ and does not change the overall bound. Space usage is $\mathcal{O}(L_1 L_2)$ for the DP table (plus a negligible $\mathcal{O}(\min\{L_1, L_2\})$ set of indices), thus $\mathcal{O}(L_1 L_2)$ overall.

### 4.4 VALIDATING ATTENTION-REASONING CORRESPONDENCE

To validate that our step-to-step attention matrices truly capture reasoning dependencies, we conduct experiments on the **Entailment Trees dataset** (Dalvi et al., 2022), which provides gold-standard reasoning dependency annotations (premise → intermediate → conclusion) for ARC (AI2 Reasoning Challenge) science exam questions. We convert examples into structured format and extract ground-truth dependencies as binary adjacency matrices. We compare the **Structured** group (feeding structured reasoning into DeepSeek-R1-Distill-Qwen-7B and measuring alignment between step-wise attention and gold dependencies) against a **Shuffled** group (randomly shuffling step order while keeping question/answer positions fixed, thus destroying reasoning structure). For each example, we compute the alignment score as the proportion of gold dependencies where attention weight exceeds the average attention, and calculate the win rate as the percentage of examples where the Structured group outperforms the Shuffled group.

As shown in Table 1, we evaluate on two reasoning scenarios from ARC questions: **Task 1 (no distractor)** contains only necessary reasoning steps, while **Task 2 (with distractor)** includes irrelevant information to test robustness. Across both tasks, the Structured group achieves significantly higher alignment with human annotations (71.27% vs 28.48% for Task 1; 72.27% vs 24.87% for Task 2) and overwhelming win rates (97.15% and 95.29%), demonstrating that attention matrices do capture meaningful reasoning dependencies rather than spurious correlations.

Table 1: Attention-dependency alignment on Entailment Trees dataset. The experimental group uses structured reasoning with preserved order, while the control group uses randomly shuffled steps.

| Task 1 no distractor | Avg Alignment | Win Rate | Task 2 with distractor | Avg Alignment | Win Rate |
|---|---|---|---|---|---|
| Shuffled Group | 28.48% | 5.50% | Shuffled Group | 24.87% | 4.71% |
| Structured Group | **71.27%** | **97.15%** | Structured Group | **72.27%** | **95.29%** |

## 5 EXPERIMENTS

In this section, we evaluate the effectiveness of the proposed structured reasoning by comparing with GRPO and other models based on the same finetuned model.

### 5.1 COMPARED EFFICIENCY, STABILITY AND EXPLAINABILITY.

For structured reasoning tuning data, we use the S1 dataset (Muennighoff et al., 2025), which contains 1,000 high-quality problems, covering science, technology, engineering and mathematics (STEM) and related domains. we select 500 high quality structured reasoning samples. In the second stage, we structured reasoning reinforcement learning on the DeepScaleR-Preview-Dataset (Luo et al., 2025b), a mathematics dataset containing 40K question-answer pairs drawn from AIME, AMC, Omni-Math (Gao et al., 2024a), and STILL (Song et al., 2025a).

**Efficiency Task.** We evaluate the effectiveness of our proposed methods by reporting Pass@1 accuracy (mean ± standard deviation) across nine benchmarks: the math (AIME 2024, AIME 2025, AMC, MATH500 (Hendrycks et al., 2021b), Minerva, Olympiad-Bench (He et al., 2024)),

reading-comprehension (DROP), law (LSAT-AR (Zhong et al., 2023)) and massive multitask (MLU-ALL-VALID (Hendrycks et al., 2021a).) using standardized evaluation protocols. For AIME24, AIME25 and AMC23, we perform evaluations in 10 seeds each, while other are evaluated in 3 seeds each. For the training of our method, the maximum response length is limited to 4k tokens, while we report 0.5k, 1k, 2k, 4k and 8k maximum token length evaluation result for efficiency display. The max-length constraint is enforced as a hard decoding cap during generation, which terminates when reaching the specified token limit or an end-of-sequence token, whichever comes first. Additionally, we compare other 1.5B models against our MaxFlow structured reasoning version across Math problems (Appendix A.6). We also performed detailed component ablation studies (Appendix B) and comparisons between LCS and MaxFlow (Appendix C).

Table 2: Benchmark Results (Pass@1 Accuracy) under different maximum response lengths. All results are reported as mean. Avg. score calculates the average across all nine benchmarks. DS is short for DeepSeek-R1. Comparison with baseline models and methods for fine-tuning 1.5B models. The shaded models are trained by otherworks. All other results are either evaluated on existing models or on models we trained using different approaches. Methods all fine-tune DeepSeek-R1-Distill-Qwen-1.5B on the same DeepScaleR dataset.

| Model | AIME'24 | AIME'25 | AMC | MATH500 | Minerva | Olympiad | DROP | LSAT-AR | MMLU-ALL | Avg. |
|---|---|---|---|---|---|---|---|---|---|---|
| **1K Maximum Response Length** | | | | | | | | | | |
| FastCuRL | 0.00 | **2.22** | 15.83 | 25.73 | 9.19 | 7.56 | 20.80 | 22.75 | 40.54 | 16.07 |
| DeepScaleR | 0.00 | 1.11 | 16.67 | 35.00 | 13.11 | 9.48 | 23.50 | 19.71 | 41.15 | 17.75 |
| DS-Distill-Qwen-1.5B | 1.11 | 1.11 | 15.83 | 27.20 | 12.01 | 8.10 | 23.25 | 24.49 | 44.05 | 17.46 |
| GRPO | 0.00 | 0.00 | **25.00** | 45.20 | 13.73 | 12.94 | 34.50 | 22.17 | 40.67 | 21.58 |
| Ours(LCS) | 1.67 | 1.11 | 22.50 | **53.40** | **20.04** | **18.04** | 32.20 | **23.15** | **44.50** | **24.28** |
| Ours(MaxFlow) | **2.22** | 1.11 | 23.33 | 44.67 | 14.95 | 14.42 | **34.65** | 22.32 | 42.04 | 22.19 |
| **2K Maximum Response Length** | | | | | | | | | | |
| FastCuRL | 1.67 | 3.33 | 27.50 | 54.90 | 17.10 | 19.41 | 28.71 | 22.75 | 45.35 | 24.52 |
| DeepScaleR | 7.78 | 5.56 | 36.67 | 65.20 | 24.63 | 27.36 | 28.70 | 22.90 | 45.13 | 29.33 |
| DS-Distill-Qwen-1.5B | 3.33 | 1.11 | 36.67 | 52.33 | 20.96 | 19.95 | 25.29 | 21.45 | 46.72 | 25.29 |
| GRPO | 6.67 | 6.67 | 45.83 | 68.07 | 24.39 | 30.52 | 38.99 | **24.04** | 45.44 | 32.33 |
| Ours(LCS) | 6.67 | **8.33** | 53.75 | **72.20** | **27.02** | **32.67** | 33.85 | 22.45 | **47.15** | **33.79** |
| Ours(MaxFlow) | 6.67 | 6.67 | 46.00 | 69.13 | 26.39 | 31.86 | **39.41** | 22.30 | 46.72 | 32.78 |
| **4K Maximum Response Length** | | | | | | | | | | |
| FastCuRL | 14.44 | 15.56 | 50.00 | 76.60 | 29.29 | 36.84 | 33.76 | 23.04 | 48.51 | 36.45 |
| DeepScaleR | 22.22 | 24.44 | 63.33 | 77.13 | 32.11 | 40.04 | 30.33 | 24.06 | 48.55 | 40.25 |
| DS-Distill-Qwen-1.5B | 14.44 | 8.89 | 47.50 | 71.73 | 29.41 | 33.58 | 25.98 | 25.07 | 47.51 | 33.79 |
| GRPO | 17.78 | 16.67 | 58.33 | 77.13 | 29.90 | 40.40 | **42.00** | 24.49 | 46.81 | 39.28 |
| Ours(LCS) | 20.67 | 18.33 | **65.00** | **78.20** | **30.51** | 41.70 | 35.25 | **25.15** | **49.65** | 40.50 |
| Ours(MaxFlow) | **27.78** | **24.44** | 60.83 | 76.73 | 29.90 | **41.90** | 40.10 | 24.59 | 48.81 | **41.68** |
| **8K Maximum Response Length** | | | | | | | | | | |
| FastCuRL | 18.89 | 17.78 | 58.33 | 78.40 | 30.50 | 42.15 | 33.00 | 23.50 | 49.51 | 39.12 |
| DeepScaleR | 36.67 | 26.67 | 77.50 | 87.80 | 33.56 | **56.22** | 33.73 | **32.17** | 48.92 | 48.14 |
| DS-Distill-Qwen-1.5B | 23.33 | 18.33 | 66.25 | 80.33 | 31.00 | 44.49 | 30.52 | 26.26 | 50.60 | 41.23 |
| GRPO | 23.33 | 21.11 | 69.17 | 83.20 | 31.37 | 48.89 | 42.23 | 24.20 | 45.98 | 43.28 |
| Ours(LCS) | 23.33 | 20.00 | 72.50 | 82.20 | 30.51 | 48.59 | 40.80 | 28.50 | **51.25** | 44.19 |
| Ours(MaxFlow) | **36.67** | **27.78** | **77.83** | **85.33** | **34.22** | 54.81 | **42.53** | 31.26 | 49.60 | **48.89** |

**Stability Task.** To assess model stability, we conduct experiments across LSAT-AR, MATH500, and Olympiad-Bench datasets (3 seeds each) under varying sampling temperatures (0.1 to 1.0). Using a fixed 8k token maximum response length, we measure accuracy variance to quantify how robust our methods are to different temperature settings, with lower variance indicating better stability.

**Explainablity Task.** We design an experiment (Interference Injection and Selective Removal, **IISR**) (A.13) to assess our ability to analyze reasoning step importance. Since existing datasets rarely provide direct importance annotations for reasoning steps, and LLM-based scoring is noisy, we inject obviously irrelevant reasoning steps into existing chains. While we cannot confirm the relative importance of original steps, we can be certain about the irrelevance of injected ones. We compare our max-flow algorithm (4.3), top-p/top-k backtracking (A.10), average step perplexity (A.10), and random selection based on their Error Filtering Efficiency (A.11) when removing 1-11 steps from mixed reasoning chains. The experiment uses 70 correctly structured reasoning examples from S1k, covering STEM and related domains, selected for their longer trace lengths and more uniform reasoning steps. We define four types of interference steps: (1) **Redundant** - statements like "Let's summarize what we've done so far, our previous work is correct" that add no value to reasoning; (2) **Distracted** - comments indicating distraction such as "This reminds me of another problem"; (3) **Harmful** - randomly injected reasoning steps from other problems; and (4) **Confused** - copies of current reasoning steps randomly injected at incorrect positions in the reasoning chain.

Table 3: Comparison with baseline models and methods for fine-tuning 7B models. Methods in the bottom section all fine-tune DeepSeek-R1-Distill-Qwen-7B on the same DeepScaleR dataset.

| Model | AIME'24 | AIME'25 | AMC | MATH500 | Minerva | Olympiad | DROP | LSAT-AR | MMLU-ALL | Avg. |
|-------|---------|---------|-----|---------|---------|----------|------|---------|----------|------|
| **1K Maximum Response Length** | | | | | | | | | | |
| Light-R1 | 3.33 | 3.33 | 25.00 | 38.33 | 17.65 | 11.95 | 43.27 | 23.04 | 54.50 | 24.49 |
| DS-Distill-Qwen-7B | 5.56 | 4.44 | 16.67 | 35.00 | 19.00 | 11.01 | 43.44 | 21.74 | 59.20 | 24.01 |
| GRPO | 8.89 | 6.67 | 27.50 | 50.73 | 23.65 | 16.69 | 49.75 | 22.03 | 56.76 | 29.19 |
| Ours(LCS) | **13.33** | 8.78 | **35.00** | **62.20** | **29.80** | **24.50** | 50.15 | **28.80** | **60.65** | **35.06** |
| Ours(MaxFlow) | 11.11 | **10.13** | 30.00 | 57.67 | 27.21 | 20.94 | **50.33** | 25.80 | 58.96 | 32.32 |
| **2K Maximum Response Length** | | | | | | | | | | |
| Light-R1 | 22.22 | 14.11 | 43.33 | 67.67 | 35.66 | 32.20 | 45.72 | 29.42 | **62.42** | 39.19 |
| DS-Distill-Qwen-7B | 15.56 | 13.33 | 38.33 | 65.33 | 32.60 | 28.89 | 45.46 | 31.09 | 63.44 | 37.11 |
| GRPO | 22.22 | 18.89 | 56.67 | 77.27 | 35.42 | 38.96 | 51.82 | 28.78 | 60.25 | 43.36 |
| Ours(LCS) | **28.89** | **25.44** | **63.50** | **80.40** | **38.90** | 42.75 | 52.90 | 31.10 | 62.20 | **47.34** |
| Ours(MaxFlow) | 26.67 | 22.22 | 60.83 | 77.53 | 37.63 | 38.87 | 51.65 | **31.13** | 61.02 | 45.28 |
| **4K Maximum Response Length** | | | | | | | | | | |
| Light-R1 | 38.89 | 35.56 | 72.50 | 80.47 | 39.34 | 48.25 | 45.72 | 34.35 | 63.49 | 50.95 |
| DS-Distill-Qwen-7B | 35.56 | 38.89 | 62.50 | 81.20 | 39.46 | 45.68 | 45.74 | 37.61 | **64.58** | 50.14 |
| GRPO | 42.22 | 35.56 | 70.00 | 84.40 | 39.09 | 50.07 | 51.79 | 37.83 | 61.45 | 52.49 |
| Ours(LCS) | 40.00 | 32.22 | 75.83 | 85.60 | 40.15 | 51.25 | **53.20** | 40.25 | 63.15 | 53.52 |
| Ours(MaxFlow) | **43.89** | **38.89** | **80.83** | **87.53** | **40.56** | **52.99** | 52.86 | **44.49** | 61.70 | **55.97** |
| **8K Maximum Response Length** | | | | | | | | | | |
| Light-R1 | 44.44 | **45.56** | 80.83 | 89.47 | 38.24 | 59.56 | 46.00 | 50.00 | 64.50 | 57.62 |
| DS-Distill-Qwen-7B | 41.11 | 42.22 | 83.50 | 91.33 | 40.69 | 59.26 | 46.07 | 52.32 | **65.97** | 58.05 |
| GRPO | 48.89 | 40.00 | 85.50 | 90.27 | 40.26 | 59.70 | 51.78 | 46.74 | 61.79 | 58.33 |
| Ours(LCS) | 46.67 | 38.89 | 82.50 | 88.80 | 39.85 | 58.45 | 52.45 | 48.90 | 62.35 | 57.65 |
| Ours(MaxFlow) | **53.78** | 41.00 | **91.25** | **92.67** | **41.54** | **61.78** | **53.03** | **52.74** | 62.77 | **61.17** |

Table 4: Comparison of model performance variance and stability under temperature changes from 0.1 to 1.0. Results on MATH500, OlympiadBench, and LSAT-AR benchmarks demonstrate model robustness across different temperature settings. Methods in the bottom section fine-tune both DeepSeek-R1-Distill-Qwen-1.5B and 7B models using the same DeepScaleR dataset.

| 1.5B Models | Performance | | | 7B Models | Performance | | |
|-------------|-------------|----|----|-----------|-------------|----|----|
| | MATH500 | OlympiadBench | LSAT-AR | | MATH500 | OlympiadBench | LSAT-AR |
| DS-Distill-Qwen-1.5B | 78.00±2.94 | 42.50±2.74 | 26.38±0.64 | DS-Distill-Qwen-7B | 91.33±1.42 | 59.26±1.44 | 52.32±2.69 |
| FastCuRL | 80.66±1.09 | 45.26±0.80 | 26.36±0.94 | Light-R1 | 89.47±1.12 | 59.56±0.57 | 50.00±3.37 |
| GRPO | 82.51±0.79 | 48.07±0.65 | 28.98±0.50 | GRPO | 90.27±0.92 | 59.70±0.70 | 46.74±2.39 |
| Ours(LCS) | 83.74±0.48 | 50.15±0.48 | 29.87±0.58 | Ours(LCS) | 90.85±0.61 | 59.72±0.62 | 49.58±2.24 |
| Ours(MaxFlow) | 85.08±**0.53** | 54.28±**0.29** | 31.35±**0.47** | Ours(MaxFlow) | 92.67±**0.48** | 61.78±**0.50** | 52.74±**2.08** |

**Models.** Our experiments are conducted on two base models: DeepSeek-R1-Distill-Qwen-1.5B and DeepSeek-R1-Distill-Qwen-7B (DeepSeek-AI et al., 2025b). For both model sizes, we train for 500 steps and derive three variants through different reinforcement learning approaches. **GRPO** represents models trained using the GRPO algorithm to optimize the reasoning process. **Max-Flow** denotes models trained with our proposed maximum flow reward, which evaluates the balance of step contributions in the reasoning response ( 4.3). **LCS** refers to models trained using a reward based on reasoning process similarity to select optimal reasoning sequences ( 4.3). The structured reasoning example can be found in A.12. The experiment comparison can be found in A.4.4.

**Baselines.** We compare our proposed methods with several state-of-the-art baselines: (1) DeepSeek-R1-Distill-Qwen-1.5B, DeepSeek-R1-Distill-Qwen-7B (DeepSeek-AI et al., 2025b); (2) FastCuRL-1.5B-Preview (Song et al., 2025b), which employs curriculum learning for reasoning; (3) DeepScaleR-1.5B (Luo et al., 2025b), which incorporates entropy regularization in GRPO; (4) GRPO (Shao et al., 2024a), which uses guided reinforcement learning for reasoning optimization; and (5) Light-R1-7B, a larger model variant. All these models, including ours, are initialized from DeepSeek-R1-Distill-1.5B/7B and subsequently fine-tuned via reinforcement learning to enhance reasoning capabilities.

**Training Details.** We train all methods (GRPO, Max-Flow reward, LCS reward) on DeepSeek-R1-Distill-Qwen-1.5B and 7B backbones. Models are initialized with a brief structured tuning pass on the 500 Structured Reasoning set (2 epochs, learning rate $1 \times 10^{-5}$, cosine schedule without floor, weight decay $1 \times 10^{-4}$, micro-batch 1 on a single A100); this stage is uniform across methods. Structure-aware optimization then uses the DeepScaleR-Preview-Dataset in bfloat16 with FlashAttention2; inference runs under vLLM (70% GPU memory, max sequence length 4096). Per-device batch size 6, gradient accumulation 4 (effective batch 24), gradient checkpointing enabled. Learning rate sweep: $\{1 \times 10^{-6}, 2 \times 10^{-6}\}$; $2 \times 10^{-6}$ works better for 1.5B, $1 \times 10^{-6}$ for 7B. Cosine scheduler with 0.1× floor; weight decay $1 \times 10^{-4}$. Each training sample yields 6 completions at temperature 0.6 (same

for evaluation). All reward settings include a Format Score (weight 1.0). The Max-Flow reward carries weight 2.0; its graph is built by thresholding step–step attention at $\tau = 0.05$ then running a capacity-scaling max flow. The LCS structural reward is length-normalized to match Format Score scale. GRPO uses $\beta = 1 \times 10^{-3}$. When a KL constraint is enabled, we set $\delta = 1 \times 10^{-4}$ (average $K \simeq 2 \times 10^{-5}$) and $\beta_{KL} = 10^3$, giving effective penalty $\beta_{KL}\delta \approx 0.1$ upon violation. No dynamic sampling or sample-inflating heuristics are used (fixed 6 candidates per prompt). Validation occurs at fixed intervals; we report the best checkpoint on held-out metrics. Structural reward computation adds only modest overhead versus the base forward and decoding costs (Appendix A.4.2).

**Efficiency Performance.** As shown in Tables 2, 3, and A.6, we evaluate all models across six mathematics-focused benchmark datasets and three out-of-domain datasets (reading, legal, and massive multitask) to demonstrate the effectiveness of MaxFlow and LCS. From Table 2, we observe that our proposed structure-aware optimization methods consistently outperform other baselines for 1.5B models. Notably, MaxFlow with 4k training length achieves significant average improvement over GRPO and surpasses DeepScaleR-1.5B-Preview, which was trained with maximum 24k length and evaluated with 32k length (Table A.6). Similarly for 7B models, Table 3 shows that the LCS method performs excellently under 4k maximum length, while MaxFlow outperforms by a large margin across the entire length range. Besides, Figure 7 shows LCS generate more correct responses in the 256-1024 token range and fewer that exceed 8192 tokens, indicating more efficient reasoning.

Table 5: Comparison of Error Filtering Efficiency (EFE) percentage when removing 3, 5, 7, and 9 steps from responses. Perplexity uses step-level lowest perplexity ordering for removal, while both top-k and MaxFlow are based on our proposed step attention matrix method.

| Method | DeepSeek-R1-Distill-1.5B | | | | DeepSeek-R1-Distill-7B | | | |
|---|---|---|---|---|---|---|---|---|
| | 3 steps | 5 steps | 7 steps | 9 steps | 3 steps | 5 steps | 7 steps | 9 steps |
| Random | 54.97 | 54.98 | 55.39 | 53.57 | 56.60 | 56.12 | 55.56 | 55.25 |
| Perplexity | 56.65 | 59.52 | 62.91 | 68.07 | 57.87 | 61.40 | 65.36 | 65.13 |
| Ours(Top-k) | 59.36 | 67.39 | 69.84 | 76.29 | 59.63 | 64.22 | 68.65 | **75.38** |
| Ours(Max-Flow) | **69.22** | **72.71** | **75.36** | **76.67** | **65.16** | **73.55** | **74.34** | 75.16 |

**Structured Reasoning Models Produce More Stable Outputs.** As shown in Table 4, we observe contrasting behaviors between baseline and structured reasoning models across temperature variations. Baseline DeepSeek-R1-Distill models exhibit significant temperature sensitivity, with performance improving substantially as temperature increases from 0.1 to 0.9. For example, the 1.5B baseline shows accuracy gains from 77.47 to 82.33 on MATH500 when temperature rises. This suggests that baseline models rely heavily on sampling diversity to achieve better performance. In contrast, our MaxFlow method maintains consistent performance across all temperature settings, achieving the lowest variance: ±0.53 on MATH500 and ±0.29 on OlympiadBench for the 1.5B model. This temperature robustness indicates that structured reasoning frameworks produce inherently stable outputs without requiring specific sampling parameters, making them more reliable.

**Structured Analysis Helps Identify Redundant Reasoning Steps.** Through IISR experiments, we found that as more reasoning steps were removed, our proposed methods based on step-matrix (See Section 4.3) (top-k, top-p, and max-flow) significantly outperformed random removal Figure A.11. The specific example can be found in Appendix A.13. Additionally, in our comparison with perplexity-based algorithms 6, we found that removing steps with the lowest PPL (PPL Bottom) performed similarly (though slightly worse) to our methods when dealing with redundant but harmless information, as such information typically has low information content and low perplexity. Interestingly, for logically confused interference, removing steps with the highest PPL (PPL Top) performed slightly better, as steps appearing in inappropriate positions caused significantly increased perplexity. This shows that PPL reflects information quantity and cannot distinguish reasoning from disruptive content. Table 5 Our step-matrix-based methods outperformed PPL-based approaches.

## 5.2 WHAT ARE THE GAINS FROM STRUCTURED REASONING MODELS?

**Benefit from Training-Free Structured Rasoning.** For large LLMs, leveraging their robust instruction-following capabilities and inherent reasoning abilities, we can effectively guide them

towards structured reasoning without additional training. Table 6 summarizes the average token lengths and accuracies across several benchmarks, including MATH500, GPQA-Diamond, MMLU-ALL-VALID, AMC23, and AIME24. Notably, the structured reasoning model achieves similar or higher accuracy with much shorter answers, e.g., on MATH500, the average reasoning token length drops from 2945 (Base) to 1577 (Structured Guidance), while accuracy remains above 92% (See Appendix A.14 for the fill in the middle prompting strategy).

Table 6: Token Length and Accuracy Analysis of DeepSeek-R1 671B Using Training-Free Structured Guidance. Results shown across Mathematical and General Benchmarks under same settings.

| Model | MATH500 Acc./Len. | GPQA Acc./Len. | MMLU Acc./Len. | AMC23 Acc./Len. |
|---|---|---|---|---|
| Base | 92.9%/2945 | 70.3%/6537 | 88.8%/989 | 100%/1716 |
| Structured Guidance | 93.0%/1577-46.5% | 71.1%/4028-38.4% | 89.6%/512-48.2% | 100%/2053+19.6% |

**Cross-Scale Emergence of Broad Mid-to-Late Step-Span Attention.** According to 70 samples from 1.5B and 7B models with our step attention matrix thresholded at 0.1, we found that layer 0 attends to an average of 6.82 reasoning steps, while layer 1 attends to only 1.41. This produces a repeating **broad-versus-local** alternation through approximately layers 0–13, suggesting an early division of labor between (i) layers that aggregate multi-step context and (ii) layers that perform local refinement anchored to the immediately preceding step. Beginning around layer 14, all subsequent layers attend to >8 steps (peaking at 12.06), marking a transition to a stable broad-span integration regime that more faithfully ranks step importance (Figure 3). The same qualitative pattern appears in both 1.5B and 7B models: early oscillatory specialization → mid/late sustained global integration. The 7B model shows a smoother (less jagged) broadening trajectory, whereas the 1.5B model preserves sharper alternating contrasts before converging. These consistent cross-scale dynamics imply (1) the broad-span mid–late blocks encode globally consolidating reasoning signals, and (2) pruning or distillation strategies could target redundant narrow-focus early layers or alternating pairs while preserving (or selectively enhancing) the globally integrative mid–late region.

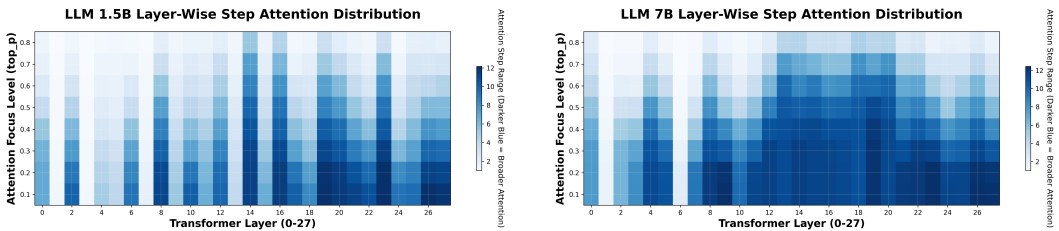

Figure 3: Analysis of attention step range: 1.5B (Left) and 7B (Right). Darker means broader.

**Analysis of Model Reasoning Patterns.** We identify a domain-invariant backbone: `assumption` → (`decompose` | `formalize`) → `verify` → `consequence` → `summarize`, while variability concentrates in domain-shaped `verify` loops (e.g., contradiction cycles in Number Theory, associative loops in Clinical, evaluative `verify`→`consequence` chains in Business Ethics, early grounding in Geometry). Algebra shows a canonical `assumption`→`decompose`→`formalize` setup, whereas Geometry's early direct formalization yields the leanest loop density and Business Ethics exhibits a high hypothesis suppression ratio with intensified evaluative chains. Loop density rises and the `verify` position shifts rightward in longer traces, signaling deferred iterative refinement. Full transition frequencies, loop densities, and positional distributions appear in Appendix A.5.

## 6 CONCLUSION

In this paper, we reformulate structured reasoning as a graph optimization problem where reasoning flows from question → steps → answer. Our approach introduces structured step annotations for reliable graph construction in small LLMs, MaxFlow reward for pruning redundant steps, LCS reward for reinforcing high-quality sub-paths. Experiments on DeepSeek-R1-Distill models demonstrate that MaxFlow provides significant performance gains with superior stability across context lengths.

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

# A APPENDIX

## A.1 THE USE OF LARGE LANGUAGE MODELS (LLMS)

As Large Language Models (LLMs) have evolved into reliable research assistance tools, we maintain transparency about their usage in this work. In accordance with the submission guidelines, we explicitly declare our LLM utilization in the following scenarios.

First, we employed LLMs for grammar and style enhancement, specifically for proofreading and improving the linguistic quality of the manuscript. All suggestions were manually reviewed and verified by the authors to ensure accuracy and appropriateness.

Second, LLMs were utilized to assist in the layout and organization of paper figures, helping optimize the arrangement and presentation of visual elements. The actual content and design decisions remained entirely under the authors' control, with LLMs providing suggestions for effective visual organization and structural composition.

We emphasize that all LLM-generated content underwent thorough human verification and refinement. The core research ideas, methodology, experiments, and conclusions were independently developed by the authors. LLMs served purely as assistive tools under careful human supervision to ensure the work's reliability and originality.

## A.2 ETHICS STATEMENT

We affirm our full compliance with the ICLR Code of Ethics throughout this research. Our work primarily focuses on visualization techniques and does not involve human subjects, sensitive personal data, or potentially harmful applications. The visualizations and methodologies presented in this paper are designed to be general-purpose tools that promote transparency and understanding in data analysis.

We acknowledge that any visualization tool could potentially be misused for misrepresenting data. To address this concern, we have implemented clear documentation of all visualization parameters, explicitly stated the limitations and appropriate use cases, and designed our tools with built-in safeguards against common forms of visual manipulation.

We declare no conflicts of interest, and our research was conducted independently without external commercial influence.

## A.3 REPRODUCIBILITY STATEMENT

We are committed to ensuring the reproducibility of our research findings. Our complete set of interactive visualizations is currently available at our anonymous website https://anonymous.4open.science/w/structured-reasoning/. These visualizations demonstrate all the key findings discussed in the paper.

While our source code and data are not yet publicly available due to the double-blind review process, we are preparing comprehensive releases that will include the complete implementation code, processing scripts and documentation, sample datasets used in our experiments, along with configuration files and parameters.

Section 3 of our paper provides detailed technical specifications and methodology, with additional implementation details available in Appendix A.

## A.4 TRAINING DETAILS AND RUNTIME ANALYSIS

We summarize the complexity of the two structural rewards: MAX-Flow and LCS, together with the scalability heuristics actually used.

### A.4.1 NOTATION

$B$ batch size; $H$ attention heads; $n$ average reasoning steps; $T_{\mathrm{avg}}$ tokens per step; $L = nT_{\mathrm{avg}}$ total reasoning length; $d_h$ head dim; $\tau$ sparsification threshold; $E$ retained edges after thresholding; $C_{\mathrm{max}}$ maximum edge capacity; $m$ number of answer candidates; $L_{\mathrm{ans}}$ average candidate length.

### A.4.2 SCALING HEURISTICS

Layer selection: extract step attention only from a small subset (e.g., 23–27 layers), cutting proportional overhead. Adaptive threshold: choose $\tau$ as a running quantile to stabilize $E$ as context length grows, avoiding quadratic blow-up. Capacity bucketing (8–12 bit) bounds $\log C_{\mathrm{max}}$ and shortens scaling phases. Structural token filtering shrinks LCS input length before any quadratic DP.

Table 7: Observed MAX-Flow reward overhead (single evaluation).

| Model | Max Len | Avg Steps | Peak Mem (MB) | Latency (ms) |
|-------|---------|-----------|---------------|--------------|
| 1.5B  | 2048    | 6.23      | 69.10         | 258          |
| 7B    | 2048    | 6.18      | 177.59        | 395          |
| 7B    | 4096    | 11.05     | 373.89        | 1321         |

### A.4.3 PRACTICAL SUMMARY

Overhead is dominated by sparse max-flow with $E \ll n^2$; LCS becomes the main cost only when doing full pairwise alignment with large $m$; layer restriction, adaptive edge sparsity, and structural token filtering preserve tractability for long contexts (e.g., 128K) without quadratic memory growth.

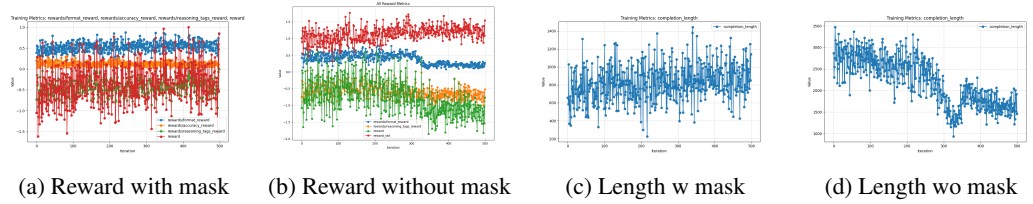

(a) Reward with mask    (b) Reward without mask    (c) Length w mask    (d) Length wo mask

Figure 4: Impact of Truncated Completion Masking on Training Stability

### A.4.4 EXPERIMENTAL DETAILS

Table 8: Training Details. To ensure consistency in counting training steps, we standardized the batch size to 128. This means that two steps with a batch size of 64 are considered equivalent to one step with a batch size of 128.

| Model | Training Steps | Training Stages | Number of GPUs Used in Each Stage |
|-------|----------------|-----------------|-----------------------------------|
| Ours | RL($\sim 23$) | 2 | 1, 4 |
| FastCuRL | RL($\sim 860$) | 4 | 8, 8, 8, 8 |
| DeepScaleR | RL($\sim 1{,}750$) | 3 | 8, 16, 32 |

**Truncation Robustness.**

Our observations reveal that truncated long outputs can induce notable gradient fluctuations and result in unstable training processes (Figure 4). To mitigate this issue, we mask the truncated completions to disregard their reward values and gradient updates. This approach effectively stabilizes optimization by omitting samples surpassing a predefined length limit.

**Tag Randomization for Robustness**

Inspired by DeepSeek-R1's reasoning completions, we introduce randomization in the order of reasoning tags in the prompt during training. Specifically, for each question, we retain the top 5 tags

and randomly sample 0–5 additional tags from the remaining set, shuffling their order in the prompt. This approach reduces overfitting to fixed reasoning patterns and encourages the model to generalize reasoning strategies.

## A.5 Domain-Specific Reasoning Patterns

**Data & Extraction.** We sample 1,000 MaxFlow reasoning traces across five domains: Algebra, Number Theory, Geometry, Clinical Knowledge, and Business Ethics. Each trace is segmented into induced step tags (e.g., `assumption`, `decompose`, `formalize`, `verify`, `case_analysis`, `association`, `consequence`, `summarize`). We construct a directed multigraph over step types; edges count adjacent transitions. Edges with global frequency $< 0.5\%$ are pruned for clarity (full graph retained for reproducibility).

**Metrics.** (1) Average step count. (2) Top tri-gram pattern (local procedural schema). (3) Verify-centered loop density: proportion of transitions incident to `verify` that participate in a back-reference to any earlier non-terminal step within a 6-step sliding window. (4) Hypothesis suppression ratio: $1 - \frac{\text{freq}(\texttt{assumption}\rightarrow\texttt{decompose})}{\text{freq}(\texttt{assumption}\rightarrow*)}$. (5) Positional distribution: normalized relative index of each tag (Fig. 8).

**Cross-Domain Findings.** A stable backbone (`assumption`→`decompose`/`formalize`→ `verify`→`consequence`→`summarize`) appears in all domains, but modulation occurs in early structural translation and verification refinement: (1) **Algebra**: canonical `assumption`→`decompose`→`formalize` pipeline before consolidation. (2) **Number Theory**: elevated `case_analysis`→ `contradiction` and `verify`↔ `contradiction` loops (proof refinement). (3) **Geometry**: suppressed `decompose`; early `assumption`→`formalize` grounding (equational or coordinate forms). (4) **Clinical**: associative diagnostic path `assumption`→`association`→`case_analysis`; verification loops link symptom clusters to differential hypotheses. (5) **Business Ethics**: sparse `assumption`→`decompose` (limited hypothesis branching), intensified evaluative `verify`→`consequence` chains and `verify`-centric loops.

**Loop Dynamics.** Loop density around `verify` increases with total step length (upper quartile traces show a right-shifted `verify` positional distribution), consistent with iterative late-stage refinement rather than premature validation. Number Theory and Business Ethics show the highest verify-loop densities (contradiction vs. evaluative implication), while Geometry exhibits the leanest loops due to early formal grounding reducing re-check cycles.

Table 9: Domain-specific structural statistics (loop density / suppression values illustrative; replace with empirical measurements).

| Domain | Avg. Steps | Top Tri-gram | Verify Loop Density |
|---|---|---|---|
| Algebra | 12.3 | assumption→decompose→formalize | 0.41 |
| Number Theory | 11.8 | case_analysis→contradiction→verify | 0.57 |
| Geometry | 10.5 | assumption→formalize→verify | 0.38 |
| Clinical | 14.2 | assumption→association→case_analysis | 0.49 |
| Business Ethics | 13.6 | verify→consequence→summarize | 0.53 |

**Implications.** The coexistence of a transferable backbone and domain-conditioned verification loops suggests: (i) pruning strategies can target high-density `verify` cycles (e.g., contradiction refinement) without harming structural progression; (ii) low hypothesis branching domains (Business Ethics) may benefit from explicit hypothesis expansion prompts; (iii) early formal grounding (Geometry) reduces downstream verification overhead—an avenue for curriculum design.

## A.6 Detailed Model Comparison and Reward Analysis

Table 11 presents a comprehensive comparison of three Small Structure Reasoning (SR) methods across various mathematical benchmarks. SR-FLOW demonstrates superior performance, achieving

Table 10: Benchmark Results (Pass@1 Accuracy). All results are reported as mean ± standard deviation. Avg. score calculates the average across all six benchmarks, while Large Avg. focuses on the more stable MATH500, Minerva, and Olympiad benchmarks. Top-3 models in each category are highlighted with increasing gray intensity.

| Model | AIME'24 | AIME'25 | AMC'23 | MATH500 | Minerva | Olympiad | Avg. | Large Avg. |
|---|---|---|---|---|---|---|---|---|
| **Based on: Qwen2.5-Math-1.5B (RL)** | | | | | | | | |
| Math | 11.3±3.6 | 5.7±2.7 | 44.0±4.9 | 51.7±5.5 | 11.3±2.2 | 26.0±0.6 | 25.0±3.3 | 29.7±2.8 |
| Oat-Zero | 16.0±3.2 | 6.7±3.4 | 52.5±2.9 | 73.5±1.7 | 26.3±0.8 | 37.2±1.3 | 32.0±2.2 | 45.7±1.3 |
| Math | 12.0±1.7 | 11.7±5.7 | 54.8±5.3 | 74.7±0.5 | 26.7±1.8 | 37.9±0.2 | 36.3±2.5 | 46.4±0.8 |
| **Based on: Deepseek-R1-Distill-Qwen-1.5B (RL)** | | | | | | | | |
| R1-Distill | 28.7±4.8 | 22.3±5.2 | 71.5±3.9 | 84.9±0.3 | 30.5±1.0 | 52.4±0.4 | 48.4±2.6 | 55.9±0.6 |
| L1-Exact | 24.4±3.3 | 22.3±4.2 | 70.5±3.7 | 86.6±0.8 | 31.5±1.7 | 52.5±1.3 | 47.9±2.5 | 56.9±1.3 |
| L1-Max | 27.7±4.2 | 21.0±5.0 | 73.2±6.0 | 84.7±0.1 | 33.3±0.9 | 52.3±0.6 | 48.7±2.8 | 56.8±0.5 |
| Open-RS1 | 28.9±6.0 | 21.3±4.2 | 75.0±3.3 | 85.1±0.8 | 30.4±0.2 | 53.2±1.9 | 49.0±2.7 | 56.2±1.0 |
| Open-RS2 | 31.3±7.7 | 22.7±5.6 | 73.0±5.7 | 84.1±0.2 | 29.2±1.1 | 53.7±0.6 | 49.0±3.5 | 55.7±0.6 |
| Open-RS3 | 29.7±4.6 | 24.7±6.5 | 69.2±5.5 | 84.2±1.1 | 28.6±2.3 | 51.8±0.8 | 48.0±3.5 | 54.9±1.4 |
| STILL-3 | 34.7±5.5 | 24.0±6.4 | 72.5±5.4 | 86.6±1.9 | 30.0±0.6 | 53.9±1.5 | 50.3±3.6 | 56.8±1.3 |
| II-Thought | 32.0±5.9 | 24.0±4.1 | 79.5±5.1 | 86.6±0.6 | 31.7±0.6 | 54.9±0.4 | 51.5±2.8 | 57.7±0.5 |
| FastCuRL | 36.3±4.3 | 27.0±3.7 | 78.8±4.1 | 87.9±1.2 | 30.8±1.4 | 56.5±0.6 | 52.9±2.6 | 58.4±1.1 |
| DeepScaleR | 37.0±6.6 | 30.3±4.3 | 76.2±4.6 | 87.8±1.0 | 31.0±1.5 | 55.5±1.1 | 53.0±3.2 | 58.1±1.2 |
| **Ours Based on: Deepseek-R1-Distill-Qwen-1.5B (RL)** | | | | | | | | |
| MAX-FLOW | 36.7±8.9 | 27.0±8.2 | 77.8±6.6 | 85.3±1.6 | 34.2±2.9 | 54.9±1.9 | 52.6±5.0 | 58.1±2.1 |

the highest average accuracy (58.1%) while requiring fewer reasoning steps. SR-LCS offers the most token-efficient approach, using approximately 20% fewer tokens while maintaining competitive accuracy. Highlighted cells indicate top performances for each benchmark and method, showing that different reasoning approaches excel in different problem domains.

Table 11: Comparative analysis of GRPO training from 0 to 500 global steps under four reward designs: accuracy (ACC), max-flow (FLOW), longest common subsequence (LCS), and their 1:1 joint combination (JOINT). We report Pass@1 accuracy (%) across benchmarks plus average reasoning steps and tokens per sample. (JOINT rows omit standard deviations: single run or variance not reported.) The table demonstrates the evolution of various metrics during GRPO training from 0 to 500 global steps under three reward functions: ACC, FLOW, LCS and JOINT. The metrics are tracked throughout the training process to show how different reward mechanisms influence the performance of the model.

| Method | Accuracy (%) | | | | | | | Steps | Tokens |
|---|---|---|---|---|---|---|---|---|---|
| | AIME24 | AIME25 | AMC23 | MATH500 | Minerva | Olympiad | Large Avg. | Avg. | Avg. |
| **GRPO** | 32.7±8.7 | 25.3±8.0 | 75.8±6.7 | 85.6±1.6 | 31.3±2.8 | 53.3±1.9 | 56.7±2.1 | 9.57 | 1873 |
| | 36.7±8.9 | 26.7±8.2 | 72.0±7.1 | 85.5±1.6 | 31.1±2.8 | 53.3±1.9 | 56.6±2.1 | 9.94 | 1796 |
| | 30.0±8.5 | 21.3±7.6 | 74.0±7.0 | 84.7±1.6 | 32.4±2.8 | 52.9±1.9 | 56.7±2.1 | 10.41 | 1808 |
| | 24.0±7.8 | 22.5±7.7 | 74.0±7.0 | 83.7±1.7 | 33.0±2.9 | 52.2±1.9 | 56.3±2.2 | 10.81 | 1859 |
| | 30.3±8.7 | 24.2±7.8 | 73.5±6.6 | 84.2±1.6 | 31.7±2.8 | 50.8±1.9 | 55.6±2.1 | 11.03 | 1828 |
| | 36.7±7.7 | 19.5±7.2 | 70.6±7.4 | 83.7±1.7 | 31.1±2.8 | 50.9±1.9 | 55.2±2.1 | 10.85 | 1854 |
| **MAX-FLOW** | 32.7±8.7 | 25.3±8.0 | 75.8±6.7 | 85.6±1.6 | 31.3±2.8 | 53.3±1.9 | 56.7±2.1 | 9.57 | 1873 |
| | 33.0±8.5 | 26.0±8.1 | 76.5±6.8 | 84.9±1.6 | 31.3±2.8 | 53.6±1.9 | 56.6±2.1 | 9.63 | 1820 |
| | 34.7±8.3 | 26.3±7.8 | 76.8±6.8 | 85.6±1.6 | 34.1±2.9 | 53.5±1.9 | 57.7±2.1 | 9.52 | 1779 |
| | 36.7±8.9 | 27.0±8.2 | 77.8±6.6 | 85.3±1.6 | 34.2±2.9 | 54.8±1.9 | 58.1±2.1 | 9.24 | 1830 |
| | 33.5±8.2 | 25.7±7.6 | 75.3±6.9 | 85.0±1.6 | 33.2±2.9 | 53.7±1.9 | 57.3±2.1 | 8.78 | 1804 |
| | 30.3±8.7 | 24.2±7.8 | 74.0±7.0 | 84.7±1.6 | 32.3±2.8 | 54.3±1.9 | 57.1±2.1 | 7.84 | 1798 |
| **LCS** | 32.7±8.7 | 25.3±8.0 | 75.8±6.7 | 85.6±1.6 | 31.3±2.8 | 53.3±1.9 | 56.7±2.1 | 9.57 | 1873 |
| | 34.0±8.7 | 24.7±7.9 | 75.0±6.9 | 84.9±1.6 | 32.0±2.8 | 54.1±1.9 | 57.0±2.1 | 9.61 | 1780 |
| | 33.3±8.7 | 22.5±7.8 | 74.0±7.0 | 84.4±1.6 | 30.9±2.8 | 51.8±1.9 | 55.7±2.1 | 10.36 | 1668 |
| | 30.3±8.5 | 23.3±7.8 | 74.0±7.0 | 83.3±1.7 | 29.9±2.8 | 51.6±1.9 | 54.9±2.1 | 10.91 | 1614 |
| | 33.7±8.7 | 23.7±7.9 | 75.0±6.9 | 84.9±1.6 | 31.8±2.8 | 50.6±1.9 | 55.8±2.1 | 11.75 | 1509 |
| | 31.0±8.5 | 23.3±7.7 | 74.8±7.1 | 84.8±1.6 | 30.5±2.8 | 52.7±1.9 | 56.0±2.1 | 11.56 | 1504 |
| **FLOW+LCS 1:1** | 33.20±8.6 | 25.15±7.9 | 75.65±6.8 | 85.45±1.6 | 31.80±2.8 | 53.55±1.9 | 56.93±1.9 | 9.72 | 1978 |
| | 33.75±8.6 | 25.60±8.0 | 75.90±6.8 | 84.75±1.6 | 31.45±2.8 | 54.10±1.9 | 57.43±1.9 | 10.38 | 1821 |
| | 33.85±8.6 | 24.65±7.9 | 75.25±6.8 | 85.20±1.6 | 32.35±2.8 | 52.90±1.9 | 57.17±1.9 | 9.53 | 1815 |
| | 33.65±8.6 | 25.40±8.0 | 76.15±6.7 | 84.15±1.6 | 32.20±2.8 | 53.45±1.9 | 57.27±1.9 | 9.91 | 1762 |
| | 33.45±8.6 | 24.85±7.9 | 75.35±6.8 | 85.10±1.6 | 32.65±2.8 | 52.40±1.9 | 57.03±1.9 | 10.67 | 1718 |
| | 30.80±8.4 | 23.90±7.8 | 74.65±6.9 | 84.90±1.6 | 31.25±2.8 | 53.75±1.9 | 57.20±2.0 | 9.65 | 1638 |

### A.7 PART OF FIGURES AND TABLES

For better layout and presentation, we have placed some figures and tables in a unified location in the Appendix.

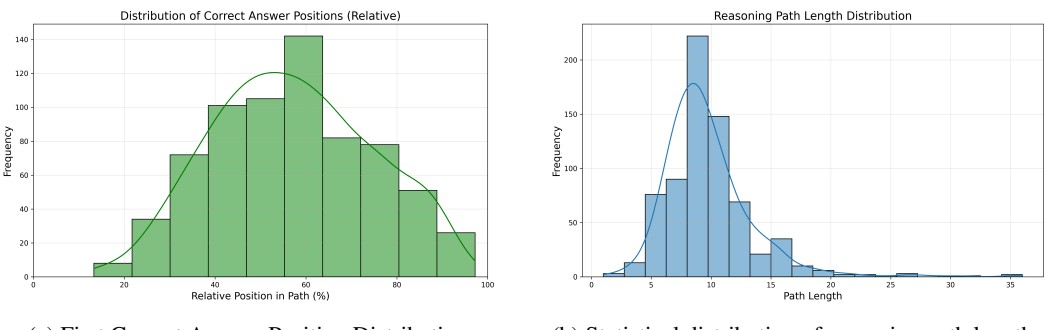

(a) First Correct Answer Position Distribution

(b) Statistical distribution of reasoning path lengths

Figure 5: Analysis of Model Reasoning Patterns: Distribution of First Correct Answers (Left) and Reasoning Path Lengths (Right).

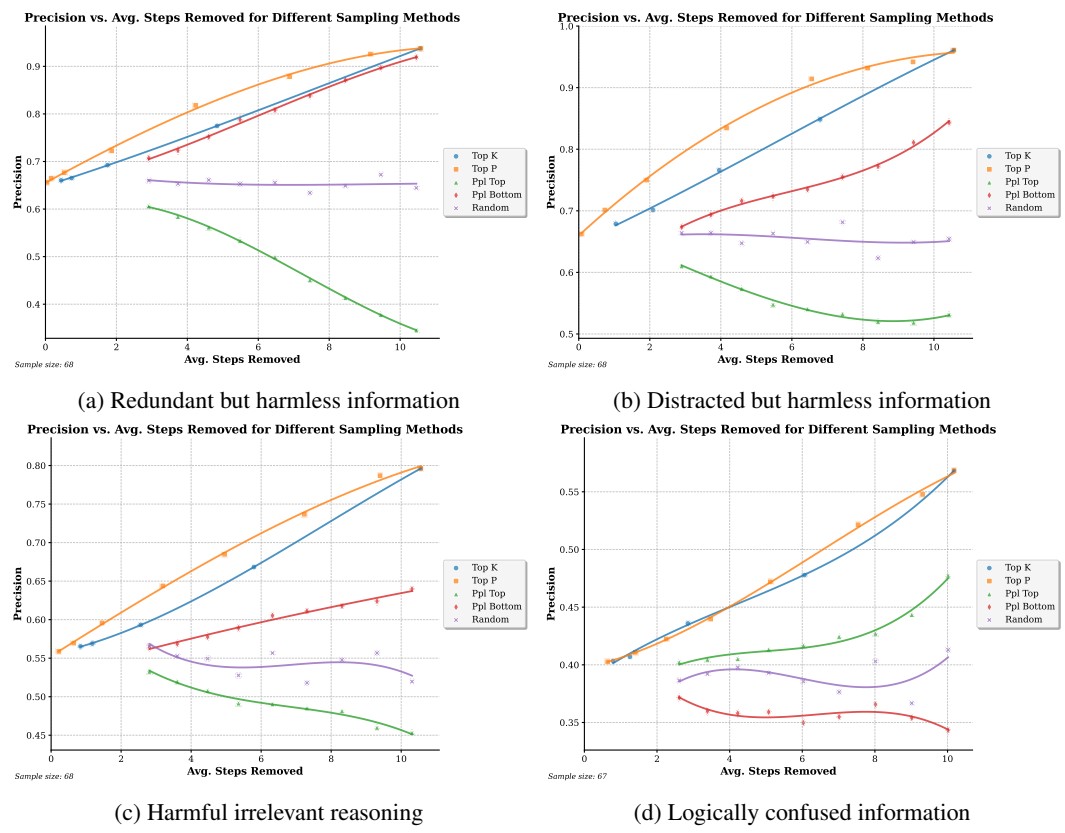

(a) Redundant but harmless information

(b) Distracted but harmless information

(c) Harmful irrelevant reasoning

(d) Logically confused information

Figure 6: IISR Results: Error Filtering Efficiency (Precision) of different algorithms when removing 1-11 steps under four types of information interference.

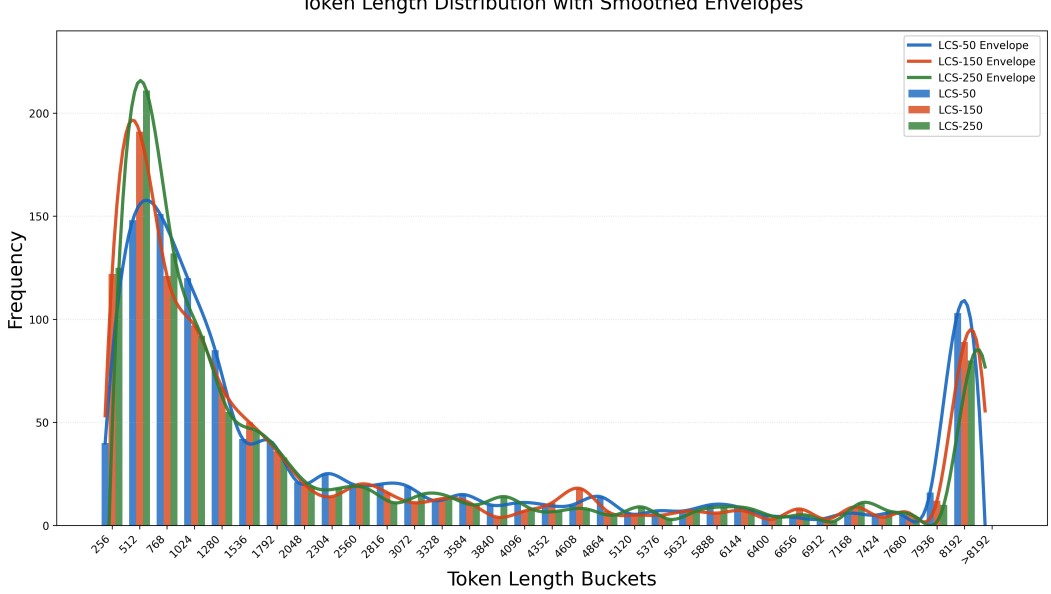

Figure 7: Token length distribution of LCS models under different training stages. The smoothed envelopes show how reward training shifts the distribution towards optimal reasoning lengths (512-1024 tokens) while maintaining performance.

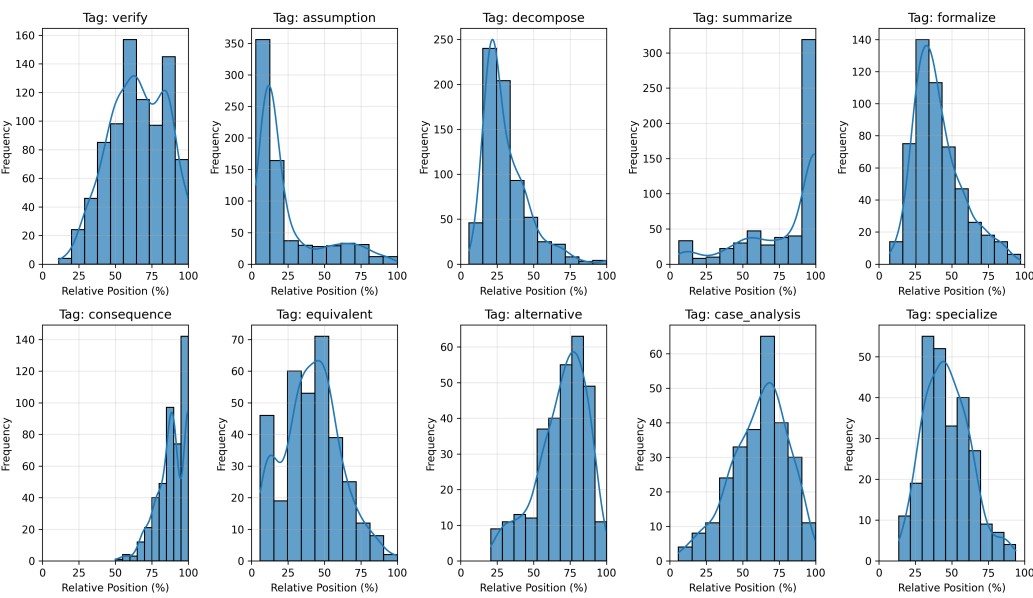

Figure 8: Distribution of the relative positions of each tag within the reasoning process.

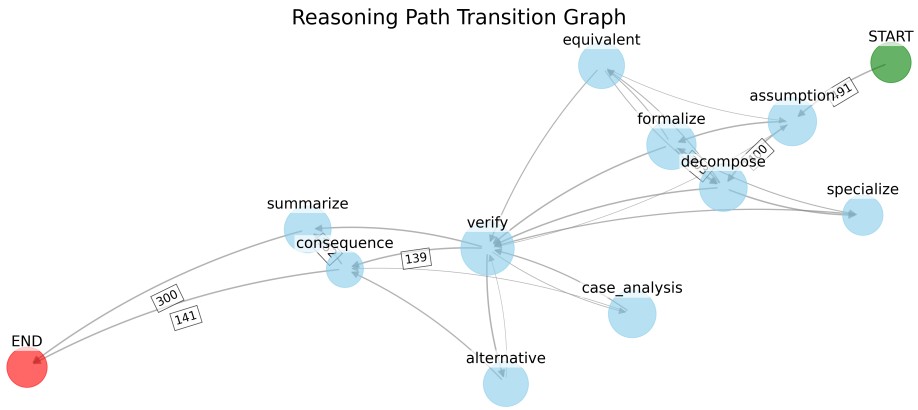

Figure 9: Illustration of Reasoning Path Transition Graph.

Table 12: Early Stopping Detection Parameters and Sample Statistics

| Metric | Value |
|---|---|
| Number of valid samples | 705 |
| Top five frequent tags | verify, summarize, equivalent, formalize, consequence |
| Token interval | 128 |
| Top five useful words | but, wait, however, check, alternatively |

## A.8 FULL PROMPTS

---

**Free Tag Chain Extraction**

Goal: Convert the raw reasoning into a linear sequence of abstract step labels (tags).
Rules: 1. Split the reasoning into semantic steps. 2. For each step invent ONE tag (a single word; use lowercase letters or underscores only; no spaces, punctuation, or digits if avoidable). 3. If two or more consecutive steps would receive the same tag, merge them into one. 4. Output ONLY one line: TAGS: tag1->tag2->tag3->...->tagK (No other text.)
Input question: {QUESTION}
Input raw reasoning: {REASONING}
Output: TAGS: ...

---

**Mathematical Problem Solving Template**

Please use the following tags at the beginning of each sentence in your reasoning: <rephrase>, <inference>, <analogy>, <equivalent>, <association>, <reverse>, <summarize>, <verify>, <complete>, <decompose>, <counterexample>, <assumption>, <constraint>, <case_analysis>, <contradiction>, <abstraction>, <formalize>, <generalize>, <specialize>, <critique>, <alternative>, <consequence>, <intuition>.
{Question}
Please reason step by step, and put your final answer within boxed{ }.

---

Table 13: Comparing Trigger Counts and Distances to First Correct Answer Across Methods.

| Trigger Type | Settings | Avg. Trigger Count ↓ | Avg. Distance to First Correct Answer (tokens) ↓ |
|---|---|---|---|
| **Top tags** | "verify", "summarize", etc. | **2.02** | **78.01** |
| Token chunks | 128-token intervals | 3.93 | 131.05 |
| Keywords | "but", "wait", "however", etc. | 2.69 | 139.97 |

---

**Multiple Choice Problem Template**

Please use the following tags at the beginning of each sentence in your reasoning: <rephrase>, <inference>, <analogy>, <equivalent>, <association>, <reverse>, <summarize>, <verify>, <complete>, <decompose>, <counterexample>, <assumption>, <constraint>, <case_analysis>, <contradiction>, <abstraction>, <formalize>, <generalize>, <specialize>, <critique>, <alternative>, <consequence>, <intuition>.
{Question}
A) {A}
B) {B}
C) {C}
D) {D}
Please reason step by step, and answer the following multiple choice question. The last line of your response should be of the following format: 'Answer: $LETTER' (without quotes) where LETTER is one of ABCD.

---

## A.9 IMPROVED COMPATIBILITY WITH TEST-TIME SCALING AND EARLY STOPPING.

For Test-time Scaling, existing work extends model outputs by injecting prompt tokens at thought-stopping points. Our method simplifies this by guiding outputs through the most likely next tag at stopping points. For early stopping, our tag-based approach outperforms traditional methods. In our experiment with 705 correct MATH500 reasoning completions (Table 12), we compared interval-based (128-token), keyword-based ("but", "wait", "however", etc.), and tag-based ("verify", "summarize", etc.) detection strategies. As Table 13 shows, our structured approach reduces average Probe-In-Middle interventions to just 2.02 while maintaining closest proximity to correct answers (78.01 tokens).

## A.10 OTHER STEP IMPORTANCE EVALUATION ALGORITHM IMPLEMENTATION

**Top-P and Top-K Selection.** Based on the step matrix computed from different layers (See Section 4.3), we implement backtracking selection methods:

$$\text{SelectSteps}(A, k, p) = \{s_i\}_{i=0}^{m}, \tag{8}$$

where $A \in \mathbb{R}^{n \times n}$ is the step attention matrix, and we select steps starting from the last step $s_{n-1}$ by either: Top-K: For each step $s_i$, select up to $k$ preceding steps with highest attention scores. Top-P: Select preceding steps with cumulative normalized attention exceeding threshold $p$.

The algorithm traverses backward from the final step, adding important preceding steps to a visited set based on attention weights, ensuring all critical reasoning dependencies are captured. **Average Perplexity.** For each step, we compute token-level perplexity:

$$\text{Perplexity}(t_i) = \frac{1}{P(t_i|x, t_1, ..., t_{i-1})}, \tag{9}$$

where $P(t_i|x, t_1, ..., t_{i-1})$ is the probability of token $t_i$ given the prompt $x$ and all preceding tokens, derived from the softmax of logits:

$$P(t_i|x, t_1, ..., t_{i-1}) = \frac{\exp(\text{logits}_i)}{\sum_j \exp(\text{logits}_j)}. \tag{10}$$

The average perplexity for a step $s$ containing tokens $\{t_1, t_2, ..., t_m\}$ is:

$$\text{AvgPerplexity}(s) = \exp\left(-\frac{1}{m}\sum_{i=1}^{m}\log P(t_i|x, t_1, ..., t_{i-1})\right). \tag{11}$$

**Random Selection.** A baseline approach where steps are selected randomly without leveraging attention patterns or perplexity metrics.

### A.11 ERROR FILTERING EFFICIENCY (EFE) EVALUATION FORMULA

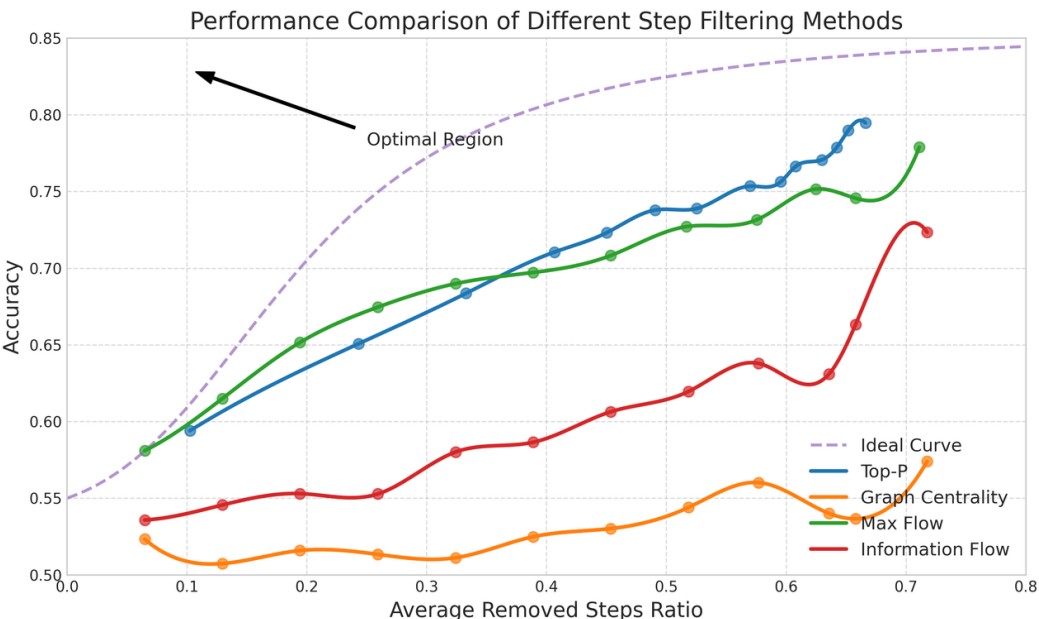

Figure 10: Comparison of Algorithms for Error Filtering Efficiency Averaged Across Four Tasks.

For the IISR experiment, where we randomly inject $N$ interference steps into an $M$-step reasoning process, the Error Filtering Efficiency is calculated as:

$$\text{EFE} = 1 - \frac{|\text{RetainedIrrelevantSteps}|}{|\text{IrrelevantSteps}|}, \tag{12}$$

where $|\text{IrrelevantSteps}|$ is the total number of interference steps injected ($N$), $|\text{RetainedIrrelevantSteps}|$ is the number of interference steps that were incorrectly retained after filtering

EFE measures the algorithm's ability to identify and remove irrelevant steps, with a value of 1.0 indicating perfect filtering (all interference steps removed) and 0.0 indicating no filtering capability.

As shown in Figure 6, we first compared Top K, Top P, Ppl Top (where higher perplexity indicates higher step importance), Ppl Bottom (the opposite), and Random. We evaluated the Error Filtering Efficiency under four different interference injection methods. The results show that Ppl-based methods exhibit unstable performance across different tasks.

As illustrated in Figure A.11, we further compared the better performing methods: Top-P, Max-Flow, and Information-Flow. We found that the Max Flow method demonstrates a superior ability in evaluating reasoning steps, particularly when removing a small number of steps.

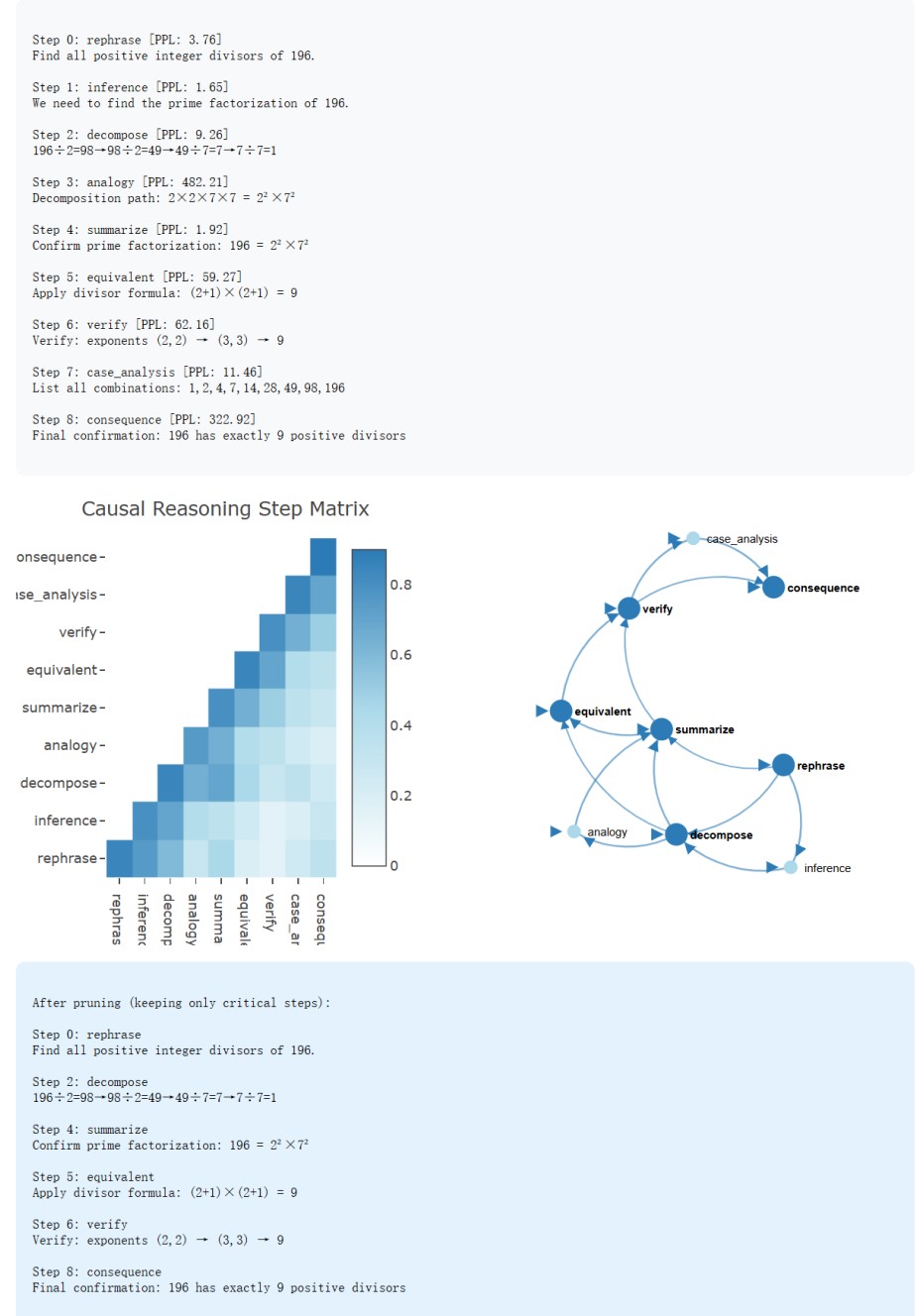

```
Step 0: rephrase [PPL: 3.76]
Find all positive integer divisors of 196.

Step 1: inference [PPL: 1.65]
We need to find the prime factorization of 196.

Step 2: decompose [PPL: 9.26]
196÷2=98→98÷2=49→49÷7=7→7÷7=1

Step 3: analogy [PPL: 482.21]
Decomposition path: 2×2×7×7 = 2²×7²

Step 4: summarize [PPL: 1.92]
Confirm prime factorization: 196 = 2²×7²

Step 5: equivalent [PPL: 59.27]
Apply divisor formula: (2+1)×(2+1) = 9

Step 6: verify [PPL: 62.16]
Verify: exponents (2,2) → (3,3) → 9

Step 7: case_analysis [PPL: 11.46]
List all combinations: 1,2,4,7,14,28,49,98,196

Step 8: consequence [PPL: 322.92]
Final confirmation: 196 has exactly 9 positive divisors
```

```
After pruning (keeping only critical steps):

Step 0: rephrase
Find all positive integer divisors of 196.

Step 2: decompose
196÷2=98→98÷2=49→49÷7=7→7÷7=1

Step 4: summarize
Confirm prime factorization: 196 = 2²×7²

Step 5: equivalent
Apply divisor formula: (2+1)×(2+1) = 9

Step 6: verify
Verify: exponents (2,2) → (3,3) → 9

Step 8: consequence
Final confirmation: 196 has exactly 9 positive divisors
```

Figure 11: Using Max-flow Method for Filtering Steps during IISR Experiment.

## A.12 STRUCTURE REASONING PROCESS DEMONSTRATION

**Input Prompt:**

Please use the following tags at the beginning of each sentence in your reasoning: <rephrase>, <inference>, <analogy>, <equivalent>, <association>, <reverse>, <summarize>, <verify>, <complete>, <decompose>, <counterexample>, <assumption>, <constraint>, <case_analysis>, <contradiction>, <abstraction>, <formalize>,

<generalize>, <specialize>, <critique>, <alternative>, <consequence>, <intuition>

Convert the point $(0, 3)$ in rectangular coordinates to polar coordinates. Enter your answer in the form $(r, \theta)$, where $r > 0$ and $0 \leq \theta < 2\pi$.

Please reason step by step, and put your final answer within ☐.

---

**<summarize>** The task is to convert the rectangular coordinates $(0, 3)$ to polar coordinates $(r, \theta)$ with $r > 0$ and $0 \leq \theta < 2\pi$.

**<assumption>** In rectangular (Cartesian) coordinates, a point is represented as $(x, y)$. To convert to polar coordinates $(r, \theta)$, we need to find:

- $r$: distance from origin
- $\theta$: angle with positive x-axis

**<formalize>** Conversion formulas:

$$r = \sqrt{x^2 + y^2}$$
$$\theta = \arctan(y/x) \quad \text{(when } x \neq 0)$$

**<decompose>** Given point $(0, 3)$:

- $x = 0$
- $y = 3$

**<specialize>** Calculate $r$:

$$r = \sqrt{0^2 + 3^2} = 3$$

**<reverse>** For $\theta$: $\arctan(3/0)$ is undefined. Observation:

- $x = 0$ and $y > 0 \Rightarrow$ point lies on positive y-axis

**<case_analysis>** Special case when $x = 0$:

$$\theta = \begin{cases} \pi/2 & \text{if } y > 0 \\ 3\pi/2 & \text{if } y < 0 \end{cases}$$

**<verify>** Check $\theta$ range: $\pi/2 \in [0, 2\pi)$ is valid.

**<complete>** Final polar coordinates:

$$\boxed{\left(3, \frac{\pi}{2}\right)}$$

---

**Output:**

The polar coordinates are $\boxed{\left(3, \dfrac{\pi}{2}\right)}$.

**Input Prompt:**

Please use the following tags at the beginning of each sentence in your reasoning: <rephrase>, <inference>, <analogy>, <equivalent>, <association>, <reverse>, <summarize>, <verify>, <complete>, <decompose>, <counterexample>, <assumption>, <constraint>, <case_analysis>, <contradiction>, <abstraction>, <formalize>, <generalize>, <specialize>, <critique>, <alternative>, <consequence>, <intuition>

What is $\sqrt{53}$ in simplest radical form?

Please reason step by step, and put your final answer within ☐.

> **\<assumption\>** To find the simplest radical form of $\sqrt{53}$, I need to check if 53 has any perfect square factors.
>
> **\<decompose\>** First, let's factorize 53 to see its prime components. Since 53 is a prime number, its only factors are 1 and 53.
>
> **\<verify\>** Since 53 is prime, there are no smaller factors other than 1 and itself. Therefore, $\sqrt{53}$ cannot be simplified further because there are no perfect squares greater than 1 that divide 53.
>
> **\<conclusion\>** Thus, the simplest radical form of $\sqrt{53}$ is $\sqrt{53}$ itself.

**Output:**

The simplest radical form is $\boxed{\sqrt{53}}$.

## A.13 EXAMPLE DEMONSTRATION OF INTERFERENCE INJECTION AND SELECTIVE REMOVAL (**IISR**)

Table 14: Steps retained by different filtering methods

| Method | Retained Steps |
|---|---|
| Top-P (p=0.7) | [1, 3, 5, 10, 11] |
| Maximum Flow | [1, 3, 5, 7, 10, 11] |
| Graph Centrality | [3, 4, 6, 8, 9, 10, 11] |
| Information Flow | [1, 2, 3, 4, 5, 6, 7, 8, 10, 11] |

- Steps marked in red: [2, 4, 6, 9]
- Interference types:
    - Irrelevant pauses (Step 2)
    - Off-topic reasoning (Steps 4)
    - Disordered repetition (Steps 6)
    - Redundant Reasoning (Step 9)

**Input Prompt:**

Please use the following tags at the beginning of each sentence in your reasoning: \<rephrase\>, \<inference\>, \<analogy\>, \<equivalent\>, \<association\>, \<reverse\>, \<summarize\>, \<verify\>, \<complete\>, \<decompose\>, \<counterexample\>, \<assumption\>, \<constraint\>, \<case_analysis\>, \<contradiction\>, \<abstraction\>, \<formalize\>, \<generalize\>, \<specialize\>, \<critique\>, \<alternative\>, \<consequence\>, \<intuition\>

What is the smallest positive perfect cube that can be written as the sum of three?

Please reason step by step, and put your final answer within $\square$.

> **\<rephrase\>** The problem is to find the smallest positive perfect cube expressible as the sum of three consecutive integers.
>
> *\<inference\> [Interference Step] Pausing to consider the implications of the next step.*
>
> **\<inference\>** First, recall that perfect cubes are numbers like 1, 8, 27. The sum of three consecutive integers can be represented as $3n$.
>
> *\<inference\> [Off-topic Interference] Gravity cancels out, so mass and distances are the key variables...*
>
> **\<analogy\>** Let the integers be $(n-1, n, n+1)$, then sum is $3n$. The smallest cube divisible by 3 is 27.

> **<inference>** *[Disordered Reasoning] Smaller cubes fail:* $1 = 1^3$ *gives non-integer* $n = \frac{1}{3}$, $8 = 2^3$ *gives* $n = \frac{8}{3}$.
>
> **<verify>** Check 27: $8 + 9 + 10 = 27$. Is there a smaller cube? No, since 1 and 8 aren't divisible by 3.
>
> **<assumption>** Alternative representation: $(n, n + 1, n + 2)$ sums to $3(n + 1)$. Still requires divisibility by 3.
>
> **<reverse>** *[Redundant Reasoning] Alternatively, maybe the three consecutive integers could be...*
>
> **<consequence>** Smaller cubes fail: $1 = 1^3$ gives non-integer $n = \frac{1}{3}$, $8 = 2^3$ gives $n = \frac{8}{3}$.
>
> **<summarize>** Therefore, the smallest positive perfect cube meeting the criteria is:
>
> $$\boxed{27}$$

## A.14 Structured reasoning through Fill In the Middle API

The full message template is structured the same as the prompt in Appendix A.8.

The API call is implemented as:

```
messages = [
    {"role": "user", "content": full_message},
    {
        "role": "assistant",
        "reasoning_content": "<rephrase>\nOkay, I will organize my thoughts
        process in a hierarchical manner.\n</rephrase>\n<",
        "content": "",
        "prefix": True
    }
]
response = await self.client.chat.completions.create(
    model=model,
    messages=messages
)
```

In particular, we initialize the reasoning process by injecting a <rephrase> tag and a metacognitive statement. This approach is effective in guiding DeepSeek-R1 to perform structured reasoning in a zero-shot setting, leading to more stable and organized reasoning patterns without additional training.

## B  Detailed Ablation Study

To better understand the contribution of the component, we conducted 9 controlled experiments to systematically evaluate the individual and combined contributions of each proposed component.

### B.1  Experimental Design

We organized our ablation experiments into two categories. The first category evaluates isolated components to measure their individual effectiveness: Structured Tags applies structured reasoning format with standard GRPO; LCS (free-form) applies LCS reward on free-form reasoning by extracting steps via \n\n separation; MaxFlow (free-form) applies MaxFlow reward on free-form reasoning with the same step extraction method; Filtered Data GRPO trains standard GRPO only on our filtered questions $\mathcal{Q}$ without any structural modifications; and Dr.GRPO implements the length-normalized GRPO variant proposed by Liu et al. (2025a) on free-form reasoning.

The second category evaluates combined approaches to understand synergistic effects: Tags + GRPO combines structured reasoning with standard GRPO; Tags + LCS combines structured reasoning with LCS reward (our proposed method); Tags + MaxFlow combines structured reasoning with MaxFlow reward (our proposed method); and Tags + Step-Level MaxFlow extends Tags + MaxFlow

Table 15: Complete ablation study showing performance improvements (percentage points) over DS-Distill-Qwen-7B baseline across different maximum response lengths. Bold indicates best performance in each column.

| Method | 1K | 2K | 4K | 8K | Average |
|---|---|---|---|---|---|
| DS-Distill-Qwen-7B (Baseline) | 0.00 | 0.00 | 0.00 | 0.00 | 0.00 |
| *Isolated Component Training* | | | | | |
|   + Structured Tags | +4.14 | +4.64 | +0.65 | +0.36 | +2.45 |
|   + LCS (free-form) | +6.23 | +4.00 | +1.00 | +0.89 | +3.03 |
|   + MaxFlow (free-form) | +3.39 | +1.25 | -0.64 | -1.09 | +0.73 |
|   + Filtered Data GRPO | +0.28 | +0.11 | +0.06 | -0.38 | +0.02 |
|   + Dr.GRPO | +4.22 | +4.97 | +1.31 | +1.37 | +2.97 |
| *Combined Component Training* | | | | | |
|   Tags + GRPO | +5.18 | +6.25 | +2.35 | +0.28 | +3.52 |
|   Tags + LCS (Ours) | **+10.79** | **+10.23** | +3.38 | -0.40 | +6.00 |
|   Tags + MaxFlow (Ours) | +8.45 | +8.17 | **+5.83** | **+3.12** | **+6.39** |
|   Tags + Step-Level MaxFlow (Ours) | +7.10 | +8.13 | +3.14 | +1.68 | +5.01 |

by applying step-level reward weighting, where each reasoning step receives importance weights normalized from MaxFlow scores (Appendix D).

All experiments use identical base models (Qwen-7B), training data, and hyperparameters. We evaluate across four maximum response length settings (1K, 2K, 4K, 8K tokens) on 9 benchmark datasets, reporting average performance improvements over the baseline DS-Distill-Qwen-7B model.

## B.2 COMPLETE ABLATION RESULTS

Table 15 presents comprehensive results across all ablation experiments. Among isolated components, structured tags alone provide +2.45% average improvement. The LCS reward on free-form reasoning achieves +3.03% average gain, showing modest effectiveness when applied to unstructured outputs. However, MaxFlow on free-form reasoning yields only +0.73% average improvement and shows negative performance at longer contexts (-0.64% at 4K, -1.09% at 8K), indicating that graph-based reward computation requires accurate step boundaries that free-form reasoning cannot reliably provide. The Dr.GRPO baseline achieves +2.97% average improvement, providing a strong comparison point for addressing GRPO's length bias.

The combined approaches demonstrate that components work better together. Tags + GRPO achieves +3.52% average, improving upon isolated structured tags (+2.45%) by an additional +1.07%. Our Tags + LCS method achieves +6.00% average improvement, performing best at shorter contexts (+10.79% at 1K, +10.23% at 2K). Our Tags + MaxFlow method achieves the highest overall performance at +6.39% average, with strongest gains at longer contexts (+5.83% at 4K, +3.12% at 8K). The step-level weighting variant (Tags + Step-Level MaxFlow) achieves +5.01% average, suggesting that assigning rewards to individual steps adds complexity without improving overall effectiveness.

## B.3 INCREMENTAL CONTRIBUTION ANALYSIS

To quantify the synergistic effects between structured reasoning and rewards, Table 16 decomposes the performance gains showing the incremental contribution of each reward method when added on top of the Structured Tags baseline.

The incremental analysis reveals that MaxFlow provides the largest additional gain (+3.95% average) when combined with structured reasoning, substantially outperforming its free-form variant which contributed only +0.73%. Similarly, LCS contributes +3.55% incremental gain on structured reasoning compared to +3.03% on free-form reasoning. This demonstrates that structured reasoning tags enable more effective reward shaping by providing accurate step boundaries for graph construction and sequence alignment.

Table 16: Incremental performance gains when adding reward methods to Structured Tags baseline (+2.45% average). Values show additional improvement beyond structured reasoning alone.

| Added Component | 1K | 2K | 4K | 8K | Avg |
|---|---|---|---|---|---|
| Structured Tags (base) | +4.14 | +4.64 | +0.65 | +0.36 | +2.45 |
| + GRPO | +1.04 | +1.61 | +1.70 | -0.08 | +1.07 |
| + LCS | +6.65 | +5.59 | +2.73 | -0.76 | +3.55 |
| + MaxFlow | +4.31 | +3.53 | +5.18 | +2.76 | **+3.95** |
| + Step-Level MaxFlow | +2.96 | +3.49 | +2.49 | +1.32 | +2.56 |

### B.4 TRAINING-FREE STRUCTURED REASONING GUIDANCE

To evaluate whether structured reasoning guidance benefits small models without additional training, we conducted experiments across three model sizes (1.5B, 7B, 14B) at different context lengths.

Table 17: Performance Comparison with and without Training Free Guidance across Different Model Sizes. AI: AIME, AMC: AMC'23, LSAT: LSAT-AR, M500: MATH500, Min.: Minerva, Oly.: OlyBench, Avg: Average.

| | With Guidance | | | | | | | | | Without Guidance | | | | | | | | |
|---|---|---|---|---|---|---|---|---|---|---|---|---|---|---|---|---|---|---|
| Tokens | AI'24 | AI'25 | AMC | LSAT | M500 | Min. | MMLU | Oly. | Avg | AI'24 | AI'25 | AMC | LSAT | M500 | Min. | MMLU | Oly. | Avg |
| | | | | | | | | | *1.5B Models* | | | | | | | | | |
| 1K | 0.00 | 0.00 | 15.83 | 19.71 | 28.33 | 11.52 | 43.91 | 8.79 | 16.01 | 1.11 | 1.11 | 15.83 | 24.49 | 27.20 | 12.01 | 44.05 | 8.10 | 16.74 |
| 2K | 3.33 | 5.56 | 31.67 | 21.59 | 52.00 | 20.83 | 47.05 | 18.62 | 25.08 | 3.33 | 1.11 | 36.67 | 21.45 | 52.33 | 20.96 | 46.72 | 19.95 | 25.32 |
| 4K | 13.33 | 13.33 | 49.17 | 22.46 | 71.73 | 26.96 | 47.72 | 33.43 | 34.77 | 14.44 | 8.89 | 47.50 | 25.07 | 71.73 | 29.41 | 47.51 | 33.58 | 34.77 |
| 8K | 23.33 | 22.22 | 68.33 | 26.38 | 81.40 | 30.64 | 47.83 | 43.60 | 42.97 | 23.33 | 18.33 | 66.25 | 26.26 | 80.33 | 31.00 | 50.60 | 44.49 | 42.57 |
| | | | | | | | | | *7B Models* | | | | | | | | | |
| 1K | 0.00 | 3.33 | 13.33 | 22.75 | 34.93 | 16.67 | 59.94 | 10.86 | 20.23 | 5.56 | 4.44 | 16.67 | 21.74 | 35.00 | 19.00 | 59.20 | 11.01 | 21.58 |
| 2K | 7.78 | 14.44 | 37.50 | 31.16 | 64.93 | 31.25 | 64.60 | 28.79 | 35.06 | 15.56 | 13.33 | 38.33 | 31.09 | 65.33 | 32.60 | 63.44 | 28.89 | 36.07 |
| 4K | 22.22 | 22.22 | 63.33 | 40.87 | 80.40 | 37.01 | 65.82 | 47.46 | 47.42 | 35.56 | 38.89 | 62.50 | 37.61 | 81.20 | 39.46 | 64.58 | 45.68 | 50.69 |
| 8K | 36.67 | 31.11 | 80.83 | 48.55 | 89.73 | 38.11 | 65.99 | 58.57 | 56.20 | 41.11 | 42.22 | 83.50 | 52.32 | 91.33 | 40.69 | 65.97 | 59.26 | 59.55 |
| | | | | | | | | | *14B Models* | | | | | | | | | |
| 4K | 33.33 | 24.44 | 65.83 | 57.10 | 84.40 | 42.28 | 82.78 | 48.99 | 54.89 | 26.67 | 23.33 | 62.50 | 55.22 | 83.00 | 39.71 | 83.87 | 47.41 | 52.71 |
| 8K | 51.17 | 34.44 | 82.67 | 72.61 | 92.10 | 43.38 | 83.02 | 61.85 | 65.16 | 50.00 | 36.67 | 82.50 | 72.17 | 91.60 | 43.38 | 84.85 | 62.81 | 65.50 |

Table 17 reveals that training-free structured reasoning guidance shows **limited and inconsistent benefits**. For the **1.5B model**, structured guidance provides minimal average improvement: +0.40% at 8K tokens. For the **7B model**, we observe negative impact. Only the **14B model** shows consistent gains: +2.18% at 4K and -0.34% at 8K. Small models lack instruction-following capabilities to utilize structured formats during inference.

## C COMPARATIVE ANALYSIS: LCS VS MAXFLOW

Both LCS and MaxFlow demonstrate strong performance when combined with structured reasoning, but exhibit distinct characteristics: LCS excels at shorter contexts (1K-2K) while MaxFlow performs better at longer contexts (4K-8K). Table 18 shows their performance converges at 3K tokens.

Table 18: Performance comparison at 3K tokens (1.5B model) showing convergence point.

| Benchmark | LCS | MaxFlow | Δ |
|---|---|---|---|
| AIME'24 | 11.33 | 11.67 | +0.34 |
| AIME'25 | 16.33 | 16.67 | +0.34 |
| AMC'23 | 61.50 | 61.25 | -0.25 |
| DROP | 38.35 | 39.51 | +1.16 |
| LSAT-AR | 25.65 | 26.30 | +0.65 |
| MATH500 | 71.10 | 73.20 | +2.10 |
| Minerva | 26.15 | 27.02 | +0.87 |
| MMLU-ALL | 44.62 | 44.84 | +0.22 |
| OlympiadBench | 37.20 | 38.44 | +1.24 |

To understand why LCS favors shorter contexts while MaxFlow excels at longer ones, we analyze response distribution patterns and reasoning metrics in Tables 19 and 20.

Table 19: Response distribution across token ranges. Numbers shown as Correct/Error. LCS concentrates correct answers in shorter ranges while MaxFlow shows balanced distribution.

| Token Range | 1.5B LCS | 1.5B MaxFlow | 7B LCS | 7B MaxFlow |
|---|---|---|---|---|
| 0-1K | 232/14 | 149/3 | 243/5 | 206/1 |
| 1K-2K | 93/8 | 158/6 | 113/7 | 133/2 |
| 2K-3K | 27/7 | 46/8 | 39/3 | 52/2 |
| 3K-4K | 12/6 | 22/6 | 21/5 | 28/3 |
| 4K-5K | 14/6 | 19/3 | 13/3 | 23/1 |
| 5K-6K | 15/4 | 13/5 | 4/3 | 12/2 |
| 6K-7K | 7/4 | 10/5 | 6/5 | 8/0 |
| 7K-8K | 9/7 | 11/5 | 5/3 | 7/1 |
| >8K (Truncated) | 0/35 | 0/31 | 0/22 | 0/19 |
| **Total** | **409/91** | **428/72** | **444/56** | **469/31** |

Table 20: Reasoning metrics comparison. LCS produces shorter steps with higher path consistency, while MaxFlow maintains flexibility with longer steps.

| Metric | 1.5B LCS | 1.5B MaxFlow | 7B LCS | 7B MaxFlow |
|---|---|---|---|---|
| Avg. Tokens per Step | 120.1 | 235.6 | 123.1 | 219.5 |
| Path Similarity (SequenceMatcher) | 0.434 | 0.410 | 0.452 | 0.423 |
| Path Similarity (Levenshtein) | 0.323 | 0.290 | 0.336 | 0.297 |
| Path Similarity (LCS Ratio) | 0.410 | 0.390 | 0.427 | 0.403 |

**LCS** operates through cross-path comparison, rewarding paths with higher common subsequence proportions. This drives the model toward shorter, more consistent reasoning steps ( 120 tokens/step vs  220 for MaxFlow) and concentrates correct answers in the 0-2K range (243 vs 206 for 7B). Higher path consistency (0.452 vs 0.423 SequenceMatcher) indicates more uniform reasoning patterns, explaining superior short-context performance.

**MaxFlow** computes flow on individual reasoning graphs, rewarding streamlined reasoning without penalizing response length. This produces more balanced answer distribution across token ranges and fewer truncated responses (19 vs 22 for 7B), resulting in better robustness at longer contexts.

## D    STEP-LEVEL REWARD IMPLEMENTATION

For Tags + Step-Level MaxFlow, we explored modulating token-level advantages using step-level importance weights derived from MaxFlow scores:

$$J_{\text{Step-GRPO}}(\theta) = \mathbb{E}\left[\frac{1}{G}\sum_{i=1}^{G}\frac{1}{|o_i|}\sum_{t=1}^{|o_i|}\min(r_{i,t}(\theta)\cdot w_{i,t}\cdot A_i, \text{clip}(r_{i,t}(\theta), 1-\varepsilon, 1+\varepsilon)\cdot w_{i,t}\cdot A_i) - \beta D_{\text{KL}}\right]$$
(13)

where $w_{i,t}$ is computed by normalizing step-level MaxFlow rewards to $[0.5, 1.5]$:

$$w_{i,t} = \begin{cases} 0.5 + \frac{R_k^{\text{MaxFlow}} - \min_j R_j}{\max_j R_j - \min_j R_j} & \text{if token } t \text{ belongs to step } k \\ 1.0 & \text{otherwise (tags, answer)} \end{cases}$$
(14)

This approach achieves +5.01% average improvement, lower than sequence-level MaxFlow (+6.39%). Fine-grained step-level credit assignment introduces complexity in determining appropriate weight scales, handling special tokens, and balancing contributions across reasoning stages. How to precisely control per-step rewards remains an open research question.

Table 21: MaxFlow Computation Time and Two-Stage Optimization Speedups. Tests conducted on dense directed graphs with varying node counts. Stage 1 (Dinic) provides $5.39\times$ average speedup, while Stage 2 (residual reuse) adds $1.38\times$ incremental improvement.

| Nodes | Tokens (n×256) | Baseline (NetworkX, s) | Optimized (Dinic, s) | Final (+Residual, s) | Speedup (Stage 1) | Speedup (Stage 2) | Total Speedup |
|---|---|---|---|---|---|---|---|
| 5 | 1,280 | 0.00046 | 0.00006 | 0.00004 | 7.96× | 1.35× | **10.76×** |
| 10 | 2,560 | 0.00087 | 0.00028 | 0.00019 | 3.14× | 1.44× | **4.53×** |
| 20 | 5,120 | 0.01766 | 0.00158 | 0.00112 | 11.16× | 1.41× | **15.76×** |
| 30 | 7,680 | 0.03044 | 0.02284 | 0.01639 | 1.33× | 1.39× | **1.86×** |
| 40 | 10,240 | 0.07580 | 0.00817 | 0.00613 | 9.28× | 1.33× | **12.36×** |
| 50 | 12,800 | 0.12053 | 0.03061 | 0.02299 | 3.94× | 1.33× | **5.24×** |
| 100 | 25,600 | 0.60409 | 0.15618 | 0.11848 | 3.87× | 1.32× | **5.10×** |
| 200 | 51,200 | 3.74300 | 0.89507 | 0.62537 | 4.18× | 1.43× | **5.99×** |
| 300 | 76,800 | 10.86789 | 2.09656 | 1.50382 | 5.18× | 1.39× | **7.23×** |
| 400 | 102,400 | 20.02902 | 4.40266 | 3.12438 | 4.55× | 1.41× | **6.41×** |
| 500 | 128,000 | 32.97774 | 6.96468 | 4.96045 | 4.73× | 1.40× | **6.22×** |

# E  IMPLEMENTATION DETAILS

## E.1  STRUCTURED DATA COLLECTION

Our structured reasoning data is collected through a four-stage pipeline. We first collect 2,000 correct free-form reasoning paths from DeepSeek-R1 on the S1K dataset. A Free Tag Chain Extraction prompt converts these raw reasoning traces into abstract step labels, yielding 23 semantically distinct tags after removing duplicates and low-frequency labels. We then combine the S1K dataset with extracted tags using a Fill-in-the-Middle API (Appendix A.14) to generate structured reasoning outputs, answering each question 8 times with randomized tag orderings. Finally, we filter based on tag coverage diversity and question difficulty, retaining 500 samples with richest tag usage and lowest correctness rates (but with at least one correct solution) as the final training set $\mathcal{Q}$.

## E.2  GRAPH CONSTRUCTION

To construct reasoning graphs from attention patterns, we aggregate token-level attention into step-wise attention matrices for each layer using the causal masking formula in Equation (3) of the main paper. Based on experiments in Section 4.3, we select layers 23-27 which focus on global reasoning patterns and compute their mean to obtain the final step-wise attention matrix $\mathbf{A}$. We designate position (0,0) as source (Question step) and position (-1,-1) as sink (Answer step), with edge weights $w_{ij} = A_{ij}$. Edges below threshold 0.05 are pruned to zero while maintaining connectivity to improve computational efficiency.

# F  MAXFLOW COMPLEXITY OPTIMIZATION

## F.1  OPTIMIZATION PIPELINE

**Stage 1: Optimized Dinic Algorithm**    We implement an efficient Dinic algorithm with level graph construction via BFS and blocking flow computation via DFS. Unlike generic max-flow solvers, our implementation exploits the structure of causal DAGs by maintaining residual capacities.

**Stage 2: Residual Network Reuse**    After computing the original max-flow, we reuse cached residual capacities from the original computation, only updating edges incident to the removed node

Algorithm 1 presents the optimized Dinic implementation, and Algorithm 2 shows the complete critical node detection pipeline with residual network reuse.

---

**Algorithm 1** Optimized Dinic Algorithm for Max-Flow Computation

---

**Require:** Graph $G = (V, E)$ with capacities $c : E \to \mathbb{R}^+$, source $s$, sink $t$
**Ensure:** Maximum flow value $f_{\max}$
 1: Initialize residual graph $G_r \leftarrow G$ with $r(u,v) \leftarrow c(u,v)$ for all $(u,v) \in E$
 2: $f_{\max} \leftarrow 0$
 3: **while** $\mathrm{BFS}(G_r, s, t)$ finds augmenting path **do**
 4:     Construct level graph $L$ via BFS from $s$
 5:     $\mathrm{level}[s] \leftarrow 0$
 6:     **for** each vertex $v$ in BFS order **do**
 7:         $\mathrm{level}[v] \leftarrow \mathrm{level}[u] + 1$ where $u$ is predecessor
 8:     **end for**
 9:     **while** $\mathrm{DFS}(s, t, \infty, L)$ finds blocking flow **do**
10:         Find augmenting path $P$ from $s$ to $t$ using DFS
11:         $\delta \leftarrow \min_{(u,v) \in P} r(u,v)$ {Bottleneck capacity}
12:         **for** each edge $(u,v) \in P$ **do**
13:             $r(u,v) \leftarrow r(u,v) - \delta$ {Update residual capacity}
14:             $r(v,u) \leftarrow r(v,u) + \delta$ {Update reverse edge}
15:         **end for**
16:         $f_{\max} \leftarrow f_{\max} + \delta$
17:     **end while**
18: **end while**
19: **return** $f_{\max}$

---

**Algorithm 2** Critical Node Detection with Residual Network Reuse

---

**Require:** Graph $G = (V, E)$, source $s$, sink $t$
**Ensure:** Set of critical nodes $\mathcal{C}$ and their contributions $\Delta_v$
 1: $f_{\mathrm{orig}} \leftarrow \mathrm{DINIC}(G, s, t)$ {Stage 1: Compute original max-flow}
 2: $\mathcal{C} \leftarrow \emptyset, \Delta \leftarrow \{\}$
 3: $V' \leftarrow V \setminus \{s, t\}$ {Candidate nodes}
 4: **for** each node $v \in V'$ **do**
 5:     {Stage 2: Fast connectivity check}
 6:     **if** $\neg\mathrm{BFS\text{-}CONNECTED}(G \setminus \{v\}, s, t)$ **then**
 7:         $\Delta_v \leftarrow f_{\mathrm{orig}}$ {Node disconnects $s$ from $t$}
 8:         $\mathcal{C} \leftarrow \mathcal{C} \cup \{v\}$
 9:         **continue**
10:     **end if**
11:     {Stage 2: Residual network reuse}
12:     Construct $G'$ by removing $v$: $E' \leftarrow \{(u,w) \in E \mid u \neq v \wedge w \neq v\}$
13:     Initialize residual graph $G'_r$ from cached residual capacities
14:     $f_{\mathrm{new}} \leftarrow \mathrm{DINIC}(G', s, t)$ {Incremental max-flow}
15:     $\Delta_v \leftarrow f_{\mathrm{orig}} - f_{\mathrm{new}}$ {Flow contribution}
16:     **if** $\Delta_v > \epsilon$ **then**
             {$\epsilon = 10^{-9}$ numerical threshold} $\mathcal{C} \leftarrow \mathcal{C} \cup \{v\}$
17:18:     **end if**
19: **end for**
20: **return** $\mathcal{C}, \Delta$

---

## F.2 Empirical Validation and Complexity Analysis

We conducted scaling tests from $n = 5$ to $n = 500$ nodes (corresponding to 128K token context with average 256 tokens per reasoning step). Table 21 demonstrates the effectiveness of both optimization stages, achieving **7.41× total speedup** over the NetworkX baseline.

Our implementation achieves empirical complexity between $\mathcal{O}(n^2 \log n)$ (R$^2$=0.9976) and $\mathcal{O}(n^{2.5})$ (R$^2$=0.9995). We therefore report the overall complexity as $\mathcal{O}(BHn^2 T_{\mathrm{avg}}) + \mathcal{O}(n^{2.5})$, where the first term (attention computation) dominates for practical reasoning chain lengths ($n \leq 500$).

