# OpenReview forum: "Structured Reasoning for LLMs: A Unified Framework for Efficiency and Explainability"
_ICLR.cc/2026/Conference — ICLR 2026 Poster_

### Official Review · Reviewer_7R1m · 2025-10-25

**Soundness:** 3
**Presentation:** 2
**Contribution:** 3
**Rating:** 6
**Confidence:** 2

**Summary:**

The paper presents a new reasoning approach, aimed at structuring the reasoning traces of LLMs.  They convert unstructured reasoning traces by prompting a teacher LLM to output a linear sequence of abstract steps (labels). After some filtering, the teacher is prompted to generate reasoning for i) a given q and ii) a set of sampled labels from the label collection.

The resulting collection is the dataset used for training the model. The latter predicts answer / reasoning from question and a "structured reasoning prompt". A graph across reasoning steps is built, where they compute the "step-to-step" attention scores to measure the  strength of an edge,

They leverage the max flow of a given node / step to assess its performance (the bigger it is, the more important to reaching the final solution). The flow analogy is used to measure the "robustness" of the solution; a higher score if the spread of importance is high, i.e. no single or few crucial nodes, but rather many pathways to the solution.

The authors evaluate on math and reasoning benchmarks. The authors show gains overall, especially when capping the response length. The authors also show that performance of the overall system is more stable to temperature variations than the baselines. Finally, the authors also demonstrate that their approach for measuring the importance of a reasoning step is the better approach when filtering out reasoning steps w.r.t. performance

**Strengths:**

1. Overall an interesting approach for structuring reasoning traces.
2. The authors' experiment protocol is quite rigorous, from the base set of results to the explainability tasks.
3. Overall good performance on the benchmarks.

**Weaknesses:**

0. The paper is very dense, from the abstract to figure 2, and even section 3.1. The overall flow of the paper could be improved, where more explanations / motivations for the equations would be appreciated, and fewer results presented. For example, what is F in equation 1 ?

1. The approach is quite cumbersome; from structuring / labelling reasoning traces to the 3 pronged loss function.

2. The paper is missing several ablations to correctly access the importance of the proposed components. Likewise, simply training on the filtered data without any additional loss terms (i.e. standard GRPO) would be very informative.

3. A comparison to Dr. GRPO [1] would be a nice addition to the paper, as they propose a simple, proper fix to the length issue in GRPO.


[1] Understanding R1-Zero-Like Training: A Critical Perspective, Liu et al., COLM 2025.

**Questions:**

1. Is a step reward given specifically to the step ? or do all step receive the same sequence level reward ?

---

> ### Author Response · Authors · 2025-11-25
> **Response to Reviewer 7R1m (1)**
>
> Thank you for your constructive comments and suggestions. We have revised our paper according to your comments. We respond to your questions below and would appreciate it if you could let us know if our response addresses your concerns.
>
> > **Q1**: The paper is very dense, from the abstract to figure 2, and even section 3.1. The overall flow of the paper could be improved, where more explanations / motivations for the equations would be appreciated, and fewer results presented. For example, what is F in equation 1 ?
>
> **A1**: Thank you for this valuable feedback. We have made revisions to improve the paper's clarity and flow:
>
> **1. Added Motivation Section:** We have introduced a dedicated Motivation section before the Method section to better explain our approach.
>
> **2. Clarified Equation 1:** $F$ refers to the filtering function $F(q,\pi,r,a)$ that we apply to verify answer correctness and reasoning difficulty. We have revised Equation 1.
>
> **3. Improved Mathematical Presentation:** following suggestions from other reviewers, we have also revised the definitions and expressions of several formulas throughout the paper (see the dark green text).
>
> >**Q2**: The approach is quite cumbersome; from structuring / labelling reasoning traces to the 3 pronged loss function.
>
> **A2**: Our design is motivated by formulating reasoning as a graph optimization problem, which naturally requires extracting structured reasoning steps. Since small LLMs struggle to reliably follow free-form prompts for step extraction, we introduce lightweight structured tags as a practical solution.
> Table R1 demonstrates that structured tags are essential for our methods to work effectively:
>
> - MaxFlow: +0.73% (w/o tags) → +3.95% (w/ tags), a +3.22% improvement
> - LCS: +3.03% (w/o tags) → +3.55% (w/ tags), a +0.52% improvement
>
> While the tags and MaxFlow introduce additional complexity, complex designs for performance gains seem common in recent work—for instance. DeepSeek-R1 combines both learned reward models (for capturing nuanced human preferences) and rule-based rewards (for math, code, and logic domains). Similarly, our structured tags effectively reduce noise in free-form reasoning, enabling MaxFlow and LCS to deliver stable improvements. We believe this complexity would be justified given the consistent performance gains and enhanced interpretability.
>
>
> **Table R1** MaxFlow and LCS Performance Gains with/without Structured Tags
>
> | Response Length | MaxFlow on Structured | MaxFlow on Free-form | LCS on Structured | LCS on Free-form |
> |----------------|---------------------:|--------------------:|------------------:|------------------:|
> | 1K | **+4.31** | +3.39 | **+6.65** | +6.23 |
> | 2K | **+3.53** | +1.25 | **+5.59** | +4.00 |
> | 4K | **+5.18** | -0.64 | **+2.73** | +1.00 |
> | 8K | **+2.76** | -1.09 | -0.76 | **+0.89** |
> | **Average** | **+3.95** | +0.73 | **+3.55** | +3.03 |

---

> ### Author Response · Authors · 2025-11-25
> **Response to Reviewer 7R1m (2)**
>
> > **Q3**: The paper is missing several ablations to correctly access the importance of the proposed components. Likewise, simply training on the filtered data without any additional loss terms (i.e. standard GRPO) would be very informative.
>
> **A3**: Following your suggestion, we conducted comprehensive ablation studies including Dr.GRPO and standard GRPO on filtered data. We performed 9 controlled experiments covering both isolated and combined components:
>
> **Isolated Component Training (5 experiments):** (1) **+ Structured Tags** – Training with tags only; (2) **+ LCS** – Training LCS on free-form reasoning model (extracting steps by `\n\n` and matching steps by similarity); (3) **+ MaxFlow** – Training MaxFlow on free-form reasoning model (extracting steps by `\n\n`); (4) **+ Filtered Data GRPO** – Training GRPO only on filtered data; (5) **+ Dr.GRPO** – Training Dr.GRPO on free-form reasoning model.
>
> **Combined Component Training (4 experiments):** (1) **Tags + GRPO** – Training GRPO on structured reasoning model; (2) **Tags + LCS** – Training LCS on structured reasoning model; (3) **Tags + MaxFlow** – Training MaxFlow on structured reasoning model; (4) **Tags + Step-Level MaxFlow** – Training MaxFlow with step-level reward on structured reasoning model.
>
> **Table R2** Performance Improvements over Baseline
>
> | Method | 1K | 2K | 4K | 8K | Average |
> |--------|----:|----:|----:|----:|--------:|
> | + Structured Tags | +4.14 | +4.64 | +0.65 | +0.36 | +2.45 |
> | + LCS | +6.23 | +4.00 | +1.00 | +0.89 | +3.03 |
> | + MaxFlow | +3.39 | +1.25 | -0.64 | -1.09 | +0.73 |
> | + Filtered Data GRPO | +0.28 | +0.11 | +0.06 | -0.38 | +0.02 |
> | + Dr.GRPO | +4.22 | +4.97 | +1.31 | +1.37 | +2.97 |
> | Tags + GRPO | +5.18 | +6.25 | +2.35 | +0.28 | +3.52 |
> | Tags + Step-Level MaxFlow | +7.10 | +8.13 | +3.14 | +1.68 | +5.01 |
> | Tags + LCS (Ours) | **+10.79** | **+10.23** | +3.38 | -0.40 | +6.00 |
> | Tags + MaxFlow (Ours) | +8.45 | +8.17 | **+5.83** | **+3.12** | **+6.39** |
>
>
> **Table R2** reveals three key findings: (1) **Filtered Data GRPO alone is insufficient** (+0.02 average) the limited data size may lead to overfitting. (2) **Dr.GRPO provides a strong baseline** (+2.97 average) by elegantly addressing GRPO's length bias. (3) **Our structured approach shows strong effectiveness**: Tags + GRPO gains **+2.45** through structured reasoning. Building on this, **Tags + LCS** achieves +6.00 (additional **+3.55** over Tags alone), and **Tags + MaxFlow** achieves +6.39 (additional **+3.95** over Tags alone), demonstrating that our reward designs effectively leverage structured reasoning. Critically, **structured tags are essential**—without them, MaxFlow only gains +0.73 due to inaccurate step extraction relying on crude `\n\n` separation, confirming that explicit structure enables our optimization methods to work effectively.
> The detailed results are presented in **Appendix B** of the revised manuscript.

---

> ### Author Response · Authors · 2025-11-25
> **Response to Reviewer 7R1m (3)**
>
> > **Q4**: Is a step reward given specifically to the step ? or do all step receive the same sequence level reward ?
>
> **A4**: Currently, we use the **same sequence-level reward** for all steps within a trajectory. Specifically, each step receives the same advantage $A_i$ derived from the final outcome reward.
>
> **Exploration of Step-Level Reward Weighting:** As shown in Table R2 and Table R3, we also explored using step-level rewards by modulating token-level advantages. Our implementation applies importance weights $w_{i,t}$ to scale the advantage for tokens belonging to different reasoning steps:
>
> $$
> J_{\text{Step-GRPO}}(\theta) = \mathbb{E} \left[ \frac{1}{G} \sum_{i=1}^G \frac{1}{|o_i|} \sum_{t=1}^{|o_i|} \left( \min(r_{i,t}(\theta) \cdot w_{i,t} \cdot A_i, \text{clip}(r_{i,t}(\theta), 1-\varepsilon, 1+\varepsilon) \cdot w_{i,t} \cdot A_i) - \beta \, D_{\text{KL}}(\pi_\theta \| \pi_{\text{ref}}) \right) \right]
> $$
>
> where $q \sim P(Q)$ and $\lbrace o_i \rbrace_{i=1}^G \sim \pi_{\theta_{\text{old}}}(O|q)$, $w_{i,t}$ is computed by normalizing step-level MaxFlow rewards to the range $[0.5, 1.5]$:
>
> $$
> w_{i,t} = \begin{cases}
> 0.5 + \frac{R_k^{\text{MaxFlow}} - \min_j R_j}{\max_j R_j - \min_j R_j} & \text{if token } t \text{ belongs to step } k \\
> 1.0 & \text{otherwise}
> \end{cases}
> $$
>
> **Empirical Results:** Step-level weighting achieves +5.01, while sequence-level reward reaches +6.39. We view fine-grained step-level reward as a promising research direction. We are actively investigating optimal reward allocation strategies, including appropriate weight scales for different reasoning stages and special token handling, to fully leverage the potential of step-level optimization.
> We have added these experimental results and discussions to **Appendix D** of the revised paper.

---

> > ### Comment · Reviewer_UN8i · 2025-11-25
> >
> > I would like to acknowledge the authors’ rebuttal, particularly the detailed clarifications provided in response to the feedback. Given the positive assessment I previously offered, I will keep the overall score unchanged.

---

### Official Review · Reviewer_UN8i · 2025-10-27

**Soundness:** 3
**Presentation:** 3
**Contribution:** 2
**Rating:** 6
**Confidence:** 3

**Summary:**

The authors propose an approach for reasoning using large language models (LLMs), introducing a framework called structured reasoning to improve efficiency through the pre-computation of the extracted common reasoning patterns. They apply this method to enhance the GRPO. The authors conduct experimental study to verify the proposed solution for insights.

**Strengths:**

This paper presents a Structured Reasoning approach that improves the efficiency, stability, and interpretability of reasoning, experimentally verified using models such as DeepSeek-R1-Distill-Qwen. The authors demonstrate that their method enhances performance even at shorter input lengths and maintains consistent quality across varied temperature settings.

A key contribution from the work is the automatic extraction of common reasoning patterns, allowing unstructured reasoning to be restructured into coherent chains via a specially designed dataset. Furthermore, the introduction of an attention-based layer-wise analysis provides detailed step-to-step attention maps, underscoring the importance of middle layers in effectively integrating reasoning context.

The authors implement two algorithms, MAX-Flow and LCS. The former creates sparse reasoning graphs to evaluate the impact of each reasoning step, and the latter identifies optimal subsequences from multiple outputs to improve reasoning quality.

Overall, the proposed methods offer promising advancements in structured reasoning frameworks.

**Weaknesses:**

The proposed three-stage pipeline for enhancing LLMs with Structured Reasoning presents an interesting and potentially valuable design that integrates multiple design choices and algorithmic components. However, this layered structure makes it challenging to clearly identify and understand the contribution and impact of each individual stage. The paper does not sufficiently analyze or disentangle these effects. Furthermore, the performance and behavior of each stage appear to depend on a number of implementation details that are not fully specified, which makes the overall approach somewhat difficult to evaluate and replicate. For example, the granularity of the labels to consider in the first step, and the edge weight definition in Step 3.

Here are a few more detailed feedback:

In Equation 1, the notation F() in the condition is not explained in the text, unless I’ve overlooked something. It seems to be a filter, but it needs to be clearly defined.

It’s great to have the complexity estimate in Section 3.3. The authors should provide some insights into whether the scale of the complexity is reasonable or not

In Equation 5, the score design heavily relies on the correct completion, which is somewhat vague in the description. As a key concept or operation, the authors should elaborate on it.

In Table 1, the performance of LCS appears stronger than MaxFlow in most cases. If this aligns with expectations, the authors should explain the rationale.

**Questions:**

see the questions in the weaknesses section.

---

> ### Author Response · Authors · 2025-11-25
> **Response to Reviewer UN8i (1)**
>
> Thank you for your constructive comments and suggestions. We have revised our paper according to your comments. We respond to your questions below and would appreciate it if you could let us know if our response addresses your concerns.
>
> > **Q1**: The proposed three-stage pipeline for enhancing LLMs with Structured Reasoning presents an interesting and potentially valuable design that integrates multiple design choices and algorithmic components. However, this layered structure makes it challenging to clearly identify and understand the contribution and impact of each individual stage. The paper does not sufficiently analyze or disentangle these effects.
>
> **A1**: In response to your suggestion and other reviewers' feedback, we conducted 9 controlled experiments covering both isolated and combined components to disentangle each component's contribution.
>
> **Isolated Component Training (5 experiments):** (1) **+ Structured Tags** – Training with tags only; (2) **+ LCS** – Training LCS on free-form reasoning model (extracting steps by `\n\n` and matching steps by similarity); (3) **+ MaxFlow** – Training MaxFlow on free-form reasoning model (extracting steps by `\n\n`); (4) **+ Filtered Data GRPO** – Training GRPO only on filtered data; (5) **+ Dr.GRPO** – Training Dr.GRPO on free-form reasoning model.
>
> **Combined Component Training (4 experiments):** (1) **Tags + GRPO** – Training GRPO on structured reasoning model; (2) **Tags + LCS** – Training LCS on structured reasoning model; (3) **Tags + MaxFlow** – Training MaxFlow on structured reasoning model; (4) **Tags + Step-Level MaxFlow** – Training MaxFlow with step-level reward on structured reasoning model.
>
> **Table R1** Performance Improvements over Baseline
>
> | Method | 1K | 2K | 4K | 8K | Average |
> |--------|----:|----:|----:|----:|--------:|
> | + Structured Tags | +4.14 | +4.64 | +0.65 | +0.36 | +2.45 |
> | + LCS | +6.23 | +4.00 | +1.00 | +0.89 | +3.03 |
> | + MaxFlow | +3.39 | +1.25 | -0.64 | -1.09 | +0.73 |
> | + Filtered Data GRPO | +0.28 | +0.11 | +0.06 | -0.38 | +0.02 |
> | + Dr.GRPO | +4.22 | +4.97 | +1.31 | +1.37 | +2.97 |
> | Tags + GRPO | +5.18 | +6.25 | +2.35 | +0.28 | +3.52 |
> | Tags + Step-Level MaxFlow | +7.10 | +8.13 | +3.14 | +1.68 | +5.01 |
> | Tags + LCS (Ours) | **+10.79** | **+10.23** | +3.38 | -0.40 | +6.00 |
> | Tags + MaxFlow (Ours) | +8.45 | +8.17 | **+5.83** | **+3.12** | **+6.39** |
>
> Table R1 reveals a clear hierarchy of contributions: Structured Tags serve as the foundation (+2.45% average), enabling accurate step-level parsing. Combined approaches achieve significantly better performance (Tags + MaxFlow: +6.39%, Tags + LCS: +6.00%), while without structured tags, both algorithms show severely limited effectiveness (MaxFlow only: +0.73%, LCS only: +3.03%). This demonstrates the necessity of the combined approach—free-form reasoning relies on crude `\n\n` separation for step extraction, which prevents MaxFlow and LCS from accurately parsing and assigning step-level rewards. Structured tags provide precise step boundaries, enabling MaxFlow to provide an additional +3.95% gain (6.39% - 2.45%) on top of structured reasoning, demonstrating the synergistic effect of our three-stage pipeline.
>
> The detailed results are presented in **Appendix B** of the revised manuscript, clearly demonstrating each component's individual contribution and their synergistic effects.

---

> ### Author Response · Authors · 2025-11-25
> **Response to Reviewer UN8i (2)**
>
> > **Q2**:  Furthermore, the performance and behavior of each stage appear to depend on a number of implementation details that are not fully specified, which makes the overall approach somewhat difficult to evaluate and replicate. For example, the granularity of the labels to consider in the first step, and the edge weight definition in Step 3.
>
> **A2**: The implementation details as follows:
>
> **1. Data Collection and Tag Consturuction**
> (1) We collected 2K FreeForm reasoning paths from correct DeepSeek-R1 answers on the S1K dataset.
> (2) These raw reasoning paths were converted into a linear sequence of abstract step labels (tags) using the Free Tag Chain Extraction prompt in (Appendix A.8).
> (3) Semantic duplicates and rare tags were removed, resulting in 23 final tags.
> (4) We then applied the Structured Reasoning Through Fill-in-the-Middle API (Appendix A.14) on the S1K dataset with tag guidance to generate stable structured reasoning outputs. Each question was answered 8 times with randomized tag order.  From the correct outputs, we filtered samples based on tag coverage and problem difficulty, retaining a subset of 500 samples with the most balanced tag usage as the final dataset.
>
> **2. Constructing the Reasoning Graph from Attention Maps**
> (1) For each layer, single-layer attention was converted into causal step-wise attention according to the formula in equation 3.
> (2) Based on our IISR experiments, we observed that middle-to-late layers capture more global reasoning, so we averaged layers $23–27$ to obtain the final step-wise attention $A$.
> (3) ​We construct the reasoning graph using the step-wise attention matrix $A$, treating $(0,0)$ corresponding to the Question as the source node and $(-1,-1)$ corresponding to the Answer as the sink node. The edge weights are set as $w_{ij}=A_{ij}$. To improve computational efficiency, edges with weights below a threshold of $0.05$ (determined empirically) are set to zero, while ensuring the graph remains connected.
>
> The detailed results are presented in **Appendix E** of the revised manuscript.
>
>
> > **Q3**:In Equation 1, the notation F() in the condition is not explained in the text, unless I’ve overlooked something. It seems to be a filter, but it needs to be clearly defined. In Equation 5, the score design heavily relies on the correct completion, which is somewhat vague in the description. As a key concept or operation, the authors should elaborate on it.
>
> **A3**: We have updated both notations in the revised manuscript:
> Equation 1: $F$ denotes a filtering function $F(q,\pi,r,a)$ that verifies the correctness of the answer and the reasoning difficulty. This has now been explicitly defined in the text near Equation 1.
> Equation 5: The LCS reward requires at least one correct completion as a reference. We have added this clarification to the description of Equation 5 to make the dependency explicit.

---

> ### Author Response · Authors · 2025-11-25
> **Response to Reviewer UN8i (3)**
>
> > **Q4**: It’s great to have the complexity estimate in Section 3.3. The authors should provide some insights into whether the scale of the complexity is reasonable or not
>
> **A4**: We have revised Section 3.3 to address this concern. We first clarify the worst-case complexity of $\mathcal{O}(n^4)$, then explicitly state that our reported empirical complexity of $\mathcal{O}(n^{2.5})$ reflects real performance. For dense attention graphs where $E = \Theta(n^2)$, our implementation employs a **two-stage optimization** to achieve this efficiency:
>
> 1. **Optimized Dinic Algorithm**: Efficient level graph construction (BFS) and blocking flow computation (DFS)
> 2. **Residual Network Reuse**: Incremental updates after node removal, caching residual capacities across queries
>
> **Empirical Validation.** We conducted scaling tests from n=5 to n=500 (128K tokens, 256 tokens/step). Table R2 shows **7.41× total speedup** over baseline, with empirical complexity between $\mathcal{O}(n^2 \log n)$ (R²=0.9976) and $\mathcal{O}(n^{2.5})$ (R²=0.9995).  We therefore report empirical complexity of $\mathcal{O}(n^{2.5})$
>
> ---
>
> **Table R2** MaxFlow Computation Time and Two-Stage Optimization Speedups
>
> | Nodes | Tokens | Baseline (NetworkX, s) | Optimized (Dinic, s) | Final (+Residual, s) | Speedup (Stage 1) | Speedup (Stage 2) | Total Speedup |
> |-------|-------------------|---------------------------|-------------------------|-------------------------|----------------------|----------------------|------------------|
> | 5     | 1,280             | 0.00046                   | 0.00006                 | 0.00004                 | 7.96×                | 1.35×                | **10.76×**       |
> | 10    | 2,560             | 0.00087                   | 0.00028                 | 0.00019                 | 3.14×                | 1.44×                | **4.53×**        |
> | 20    | 5,120             | 0.01766                   | 0.00158                 | 0.00112                 | 11.16×               | 1.41×                | **15.76×**       |
> | 30    | 7,680             | 0.03044                   | 0.02284                 | 0.01639                 | 1.33×                | 1.39×                | **1.86×**        |
> | 40    | 10,240            | 0.07580                   | 0.00817                 | 0.00613                 | 9.28×                | 1.33×                | **12.36×**       |
> | 50    | 12,800            | 0.12053                   | 0.03061                 | 0.02299                 | 3.94×                | 1.33×                | **5.24×**        |
> | 100   | 25,600            | 0.60409                   | 0.15618                 | 0.11848                 | 3.87×                | 1.32×                | **5.10×**        |
> | 200   | 51,200            | 3.74300                   | 0.89507                 | 0.62537                 | 4.18×                | 1.43×                | **5.99×**        |
> | 300   | 76,800            | 10.86789                  | 2.09656                 | 1.50382                 | 5.18×                | 1.39×                | **7.23×**        |
> | 400   | 102,400           | 20.02902                  | 4.40266                 | 3.12438                 | 4.55×                | 1.41×                | **6.41×**        |
> | 500   | 128,000           | 32.97774                  | 6.96468                 | 4.96045                 | 4.73×                | 1.40×                | **6.65×**        |
>
> *Average speedup: Stage 1 = 5.39×, Stage 2 = 1.38× (range: 1.31×-1.44×), Total = 7.41×*
>
> ---
>
> We have updated the paper to clarify these optimizations, and detailed algorithm pseudocode is provided in **Appendix F**.

---

> ### Author Response · Authors · 2025-11-25
> **Response to Reviewer UN8i (4)**
>
> > **Q5**: In Table 1, the performance of LCS appears stronger than MaxFlow in most cases. If this aligns with expectations, the authors should explain the rationale.
>
> **A5**: Table R3 shows LCS and MaxFlow perform comparably at 3K tokens, with LCS stronger on 1–3K and MaxFlow stronger on 3K–8K. Notably, MaxFlow consistently outperforms GRPO across all context lengths (1K–8K).
>
> **Table R3** 1.5B Model Performance Comparison (3072 tokens configuration)
> | Dataset           | 1.5B LCS | 1.5B MaxFlow |     Δ |
> | :---------------- | -------: | -----------: | ----: |
> | **AIME'24**       |    11.33 |        **11.67** | +0.34 |
> | **AIME'25**       |    16.33 |        **16.67** | +0.34 |
> | **AMC'23**        |    **61.50** |        61.25 | -0.25 |
> | **DROP**          |    38.35 |        **39.51** | +1.16 |
> | **LSAT-AR**       |    25.65 |        **26.30** | +0.65 |
> | **MATH500**       |    71.10 |        **73.20** | +2.10 |
> | **Minerva**       |    26.15 |        **27.02** | +0.87 |
> | **MMLU-ALL**      |    44.62 |        **44.84** | +0.22 |
> | **OlympiadBench** |    37.20 |        **38.44** | +1.24 |
>
>
> Why Lcs shorter better and MaxFlow longer better?
>
> To understand these differences, we compare the response distribution (Table R4) and reasoning metrics (Table R5).
>
> LCS operates through cross-path comparison: rewarding paths with higher common subsequence proportions and more concise step formulations. This mechanism leads LCS to:
> - concentrates correct answers in the 0–2K token range(243 vs. 206 for 7B)
> - produces shorter reasoning steps (~120 vs. ~220 tokens per step)
> - achieves higher path consistency (0.452 vs. 0.423 SequenceMatcher)
>
> This dual-level optimization drives the model toward shorter, more consistent responses, explaining LCS's superior performance on short-context tasks.
>
> MaxFlow reward to encourage streamlined reasoning chains where all steps contribute meaningfully to the final answer. This mechanism leads MaxFlow to:
> - remain robust at longer contexts (4K-8K tokens)
> - distribute correct answers more evenly across token ranges, with fewer truncated responses (19 vs. 22 for 7B)
> - maintain more stable error rates across token ranges
>
> This mechanism rewards streamlined reasoning while maintaining flexibility in response length, explaining MaxFlow’s superior performance on long-context tasks.
> We have added these experimental results and discussions to **Appendix C** of the revised paper.
>
> **Table R4** Comparison of Response Distributions Between LCS and MaxFlow
>
> | Token Range | 1.5B LCS | 1.5B MaxFlow | 7B LCS | 7B MaxFlow |
> |-------------|:--------:|:------------:|:------:|:----------:|
> | 0-1K | 232 / 14 | 149 / 3 | 243 / 5 | 206 / 1 |
> | 1K-2K | 93 / 8 | 158 / 6 | 113 / 7 | 133 / 2 |
> | 2K-3K | 27 / 7 | 46 / 8 | 39 / 3 | 52 / 2 |
> | 3K-4K | 12 / 6 | 22 / 6 | 21 / 5 | 28 / 3 |
> | 4K-5K | 14 / 6 | 19 / 3 | 13 / 3 | 23 / 1 |
> | 5K-6K | 15 / 4 | 13 / 5 | 4 / 3 | 12 / 2 |
> | 6K-7K | 7 / 4 | 10 / 5 | 6 / 5 | 8 / 0 |
> | 7K-8K | 9 / 7 | 11 / 5 | 5 / 3 | 7 / 1 |
> | >8K (Truncated) | 0 / 35 | 0 / 31 | 0 / 22 | 0 / 19 |
> | **Total** | **409 / 91** | **428 / 72** | **444 / 56** | **469 / 31** |
>
> *Note: Numbers shown as Correct / Error*
>
> **Table R5** Reasoning Step Length and Path Consistency Metrics
>
> | Metric | 1.5B LCS | 1.5B MaxFlow | 7B LCS | 7B MaxFlow |
> |--------|:--------:|:------------:|:------:|:----------:|
> | **Avg. Tokens per Step** | 120.1 | 235.6 | 123.1 | 219.5 |
> | **Path Similarity (SequenceMatcher)** | 0.434 | 0.410 | 0.452 | 0.423 |
> | **Path Similarity (Levenshtein)** | 0.323 | 0.290 | 0.336 | 0.297 |
> | **Path Similarity (LCS Ratio)** | 0.410 | 0.390 | 0.427 | 0.403 |

---

> > ### Comment · Reviewer_UN8i · 2025-11-25
> >
> > I would like to acknowledge the authors’ rebuttal, particularly the detailed clarifications provided in response to the feedback. Given the positive assessment I previously offered, I will keep the overall score unchanged.

---

> ### Author Response · Authors · 2025-11-28
>
> Dear Reviewer UN8i,
>
> We sincerely thank you for your detailed feedback, we are happy to see that our response address your concerns. Thanks again for your valuable comments to help us improve our paper.

---

### Official Review · Reviewer_sQTy · 2025-10-31

**Soundness:** 2
**Presentation:** 3
**Contribution:** 2
**Rating:** 4
**Confidence:** 4

**Summary:**

This work targets the limitations of LLM on complex reasoning tasks such as logical deduction and planning, with a focus on efficiency and explaninability.
It aims to enhance the reasoning capabilities of LLM at the step-level by introducing the structured reasoning approach, viewing the reasoning process as a directed graph.
The proposed framework first extracts the unstructured data into a structured format by incorporating the explicit reasoning steps, which enable adaptive fine-tuning of LLM afterward.
The authors also propose a layer-wise dependency tracing procedure using step-to-step attention matrices for in-depth analysis.
The fine-tuning part, they also extend GRPO with two structure-aware algorithms- Max-Flow and Longest Common Subsequence.

The empirical experiments showcase the improved efficacy, efficiency as well as the stability.

**Strengths:**

1. This paper is well-motivated, and the proposed techniques are rational.
2. The paper is well-written and easy to follow.
3. The experiments are comprehensive, covering efficacy, efficiency, and stability.

**Weaknesses:**

1. As extensions of GRPO, the two proposed variants do not always demonstrate clear advantages over GRPO. For example, in some cases, MaxFlow outperforms GRPO while LCS achieves similar performance; in other cases, LCS performs better, whereas MaxFlow's results are comparable to GRPO.
However, there is no discussion explaining these observations or directly comparing LCS and MaxFlow.

2. The stability and efficiency study only considers the MaxFlow variant while the LCS variant is ignored.

**Questions:**

1. Can smaller LLMs also benefit from the training-free structured reasoning?

---

> ### Author Response · Authors · 2025-11-25
> **Response to Reviewer sQTy (1)**
>
> Thank you for your comments and suggestions. We have revised our paper according to your comments. We respond to your questions below and would appreciate it if you could let us know if our response addresses your concerns.
>
> > **Q1**: As extensions of GRPO, the two proposed variants do not always demonstrate clear advantages over GRPO. For example, in some cases, MaxFlow outperforms GRPO while LCS achieves similar performance; in other cases, LCS performs better, whereas MaxFlow's results are comparable to GRPO. However, there is no discussion explaining these observations or directly comparing LCS and MaxFlow.
>
> **A1.1**: Performance Advantages over GRPO
>
> We respectfully clarify that there may be a misunderstanding regarding the baseline comparison.
> As shown in Table R1, MaxFlow demonstrate clear advantages over GRPO with Structured Tag Training across all context lengths (1K-8K tokens) and LCS significantly outperforms GRPO at shorter contexts (1K-2K tokens).
>
> **Table R1** Performance (7B) Gains of LCS and MaxFlow over Tags + GRPO
>
> | Method | 1K | 2K | 4K | 8K | Average |
> |--------|----:|----:|----:|----:|--------:|
> | Tags + GRPO (Baseline) | 0.00 | 0.00 | 0.00 | 0.00 | 0.00 |
> | Tags + LCS (Ours) | **+5.61** | **+3.98** | +1.03 | -0.68 | +2.48 |
> | Tags + MaxFlow (Ours) | +3.27 | +1.92 | **+3.48** | **+2.84** | **+2.87** |
>
> **A1.2**: Why LCS excels at short contexts while MaxFlow performs better at long contexts ?
> To understand these differences, we compare the response distribution (Table R2) and reasoning metrics (Table R3).
>
> LCS operates through cross-path comparison: rewarding paths with higher common subsequence proportions and more concise step formulations. This mechanism leads LCS to:
> - produces shorter reasoning steps (~120 vs. ~220 tokens per step)
> - achieves higher path consistency (0.452 vs. 0.423 SequenceMatcher)
> - concentrates correct answers in the 0–2K token range(243 vs. 206 for 7B)
>
> This dual-level optimization drives the model toward shorter, more consistent responses, explaining LCS's better performance on short-context tasks.
>
> MaxFlow reward to encourage streamlined reasoning chains where all steps contribute meaningfully to the final answer. This mechanism leads MaxFlow to:
> - remain robust at longer contexts (4K-8K tokens)
> - distribute correct answers more evenly across token ranges, with fewer truncated responses (19 vs. 22 for 7B)
> - maintain more stable error rates across token ranges
>
> This mechanism rewards streamlined reasoning while maintaining flexibility in response length, explaining MaxFlow’s better performance on long-context tasks.
> We have added these experimental results and discussions to **Appendix C** of the revised paper.
>
> **Table R2** Comparison of Response Distributions Between LCS and MaxFlow
> | Token Range | 1.5B LCS | 1.5B MaxFlow | 7B LCS | 7B MaxFlow |
> |-------------|:--------:|:------------:|:------:|:----------:|
> | 0-1K | 232 / 14 | 149 / 3 | 243 / 5 | 206 / 1 |
> | 1K-2K | 93 / 8 | 158 / 6 | 113 / 7 | 133 / 2 |
> | 2K-3K | 27 / 7 | 46 / 8 | 39 / 3 | 52 / 2 |
> | 3K-4K | 12 / 6 | 22 / 6 | 21 / 5 | 28 / 3 |
> | 4K-5K | 14 / 6 | 19 / 3 | 13 / 3 | 23 / 1 |
> | 5K-6K | 15 / 4 | 13 / 5 | 4 / 3 | 12 / 2 |
> | 6K-7K | 7 / 4 | 10 / 5 | 6 / 5 | 8 / 0 |
> | 7K-8K | 9 / 7 | 11 / 5 | 5 / 3 | 7 / 1 |
> | >8K (Truncated) | 0 / 35 | 0 / 31 | 0 / 22 | 0 / 19 |
> | **Total** | **409 / 91** | **428 / 72** | **444 / 56** | **469 / 31** |
> *Note: Numbers shown as Correct / Error*
>
>
> **Table R3** Reasoning Step Length and Path Consistency Metrics
> | Metric | 1.5B LCS | 1.5B MaxFlow | 7B LCS | 7B MaxFlow |
> |--------|:--------:|:------------:|:------:|:----------:|
> | **Avg. Tokens per Step** | 120.1 | 235.6 | 123.1 | 219.5 |
> | **Path Similarity (SequenceMatcher)** | 0.434 | 0.410 | 0.452 | 0.423 |
> | **Path Similarity (Levenshtein)** | 0.323 | 0.290 | 0.336 | 0.297 |
> | **Path Similarity (LCS Ratio)** | 0.410 | 0.390 | 0.427 | 0.403 |

---

> ### Author Response · Authors · 2025-11-25
> **Response to Reviewer sQTy (2)**
>
> > **Q2:** The stability and efficiency study only considers the MaxFlow variant while the LCS variant is ignored.
>
> **A2**: For stability analysis, Table R4 demonstrates that LCS also exhibits stable performance across different temperature settings, with standard deviations comparable to or lower than baseline methods. The efficiency results for both LCS and MaxFlow are presented in Table 2 and Table 3 of the main paper. The complete stability results have been added to **Table 4** in the revised manuscript.
>
> **Table R4** Stability Analysis of LCS and MaxFlow Methods
> | 1.5B Models          |    MATH500     | OlympiadBench  |    LSAT-AR     | 7B Models          |    MATH500     | OlympiadBench  |    LSAT-AR     |
> | :------------------- | :------------: | :------------: | :------------: | :----------------- | :------------: | :------------: | :------------: |
> | DS-Distill-Qwen-1.5B |   78.00±2.94   |   42.50±2.74   |   26.38±0.64   | DS-Distill-Qwen-7B |   91.33±1.42   |   59.26±1.44   |   52.32±2.69   |
> | FastCuRL             |   80.66±1.09   |   45.26±0.80   |   26.36±0.94   | Light-R1           |   89.47±1.12   |   59.56±0.57   |   50.00±3.37   |
> | **Ours(LCS)**                  |   83.74±**0.48**   |   50.15±0.48   |   29.87±0.58   | **Ours(LCS)**                |   90.85±0.61   |   59.72±0.62   |   49.58±2.24   |
> | **Ours(MaxFlow)**    | 85.08±0.53 | 54.28±**0.29** | 31.35±**0.47** | **Ours(MaxFlow)**  | 92.67±**0.48** | 61.78±**0.50** | 52.74±**2.08** |
>
> > **Q3:** Can smaller LLMs also benefit from the training-free structured reasoning?
>
> **A3**: Smaller LLMs show limited or inconsistent benefits from training-free structured reasoning (for 1.5B models, +0.4% at 8K tokens), whereas larger models, such as the 14B model, show more notable gains (+2.2% at 4K tokens). This limited effectiveness for smaller LLMs may be attributed to their weaker ability to follow structured reasoning instructions during reasoning.
>
> **Table R5** Scaling Experiment Results (4K & 8K tokens)
> | Model Size | Guidance | AIME'24 | AIME'25 | AMC'23 | LSAT-AR | MATH500 | Minerva | MMLU-ALL | OlympiadBench | Avg.  |
> | :--------- | :------- | ------: | ------: | -----: | ------: | ------: | ------: | -------: | ------------: | ----: |
> | **1.5B**   | 4K w/    |   13.33 |   **13.33** |  **49.17** |   22.46 |   71.73 |   26.96 |    **47.72** |         33.43 | 34.77 |
> | **1.5B**   | 4K w/o   |   **14.44** |    8.89 |  47.50 |   **25.07** |   71.73 |   **29.41** |    47.51 |         **33.58** | 34.77 |
> | **1.5B**   | 8K w/    |   23.33 |   **22.22** |  **68.33** |   **26.38** |   **81.40** |   30.64 |    47.83 |         43.60 | **42.97** |
> | **1.5B**   | 8K w/o   |   23.33 |   18.33 |  66.25 |   26.26 |   80.33 |   **31.00** |    **50.60** |         **44.49** | 42.57 |
> | **7B**     | 4K w/    |   22.22 |   22.22 |  **63.33** |   **40.87** |   80.40 |   37.01 |    **65.82** |         **47.46** | 47.42 |
> | **7B**     | 4K w/o   |   **35.56** |   **38.89** |  62.50 |   37.61 |   **81.20** |   **39.46** |    64.58 |         45.68 | **50.69** |
> | **7B**     | 8K w/    |   36.67 |   31.11 |   80.83 |   48.55 |   89.73 |   38.11 |    **65.99** |         58.57 | 56.20 |
> | **7B**     | 8K w/o   |   **41.11** |   **42.22** |  **83.50** |   **52.32** |   **91.33** |   **40.69** |    65.97 |         **59.26** | **59.55** |
> | **14B**    | 4K w/    |   **33.33** |   **24.44** |  **65.83** |   **57.10** |   **84.40** |   **42.28** |    82.78 |         **48.99** | **54.89** |
> | **14B**    | 4K w/o   |   26.67 |   23.33 |  62.50 |   55.22 |   83.00 |   39.71 |    **83.87** |         47.41 | 52.71 |
> | **14B**    | 8K w/    |   **51.17** |   34.44 |  **82.67** |   **72.61** |   **92.10** |   43.38 |    83.02 |         61.85 | 65.16 |
> | **14B**    | 8K w/o   |   50.00 |   **36.67** |  82.50 |   72.17 |   91.60 |   43.38 |    **84.85** |         **62.81** | **65.50** |
>
>
> The detail results have been added to **Appendix B.4** in the revised manuscript.

---

> > ### Comment · Reviewer_sQTy · 2025-11-26
> >
> > Thank you for the detailed response.
> >
> > However, I remain concerned regarding the results in Table 2; while the average demonstrates an overall improvement, the performance is inconsistent across individual datasets.
> > For example, on 4K/8K maximum response length, GRPO is better than LCS on the DROP dataset.
> >
> > However, it is understandable that it is not realistic to expect improvement in all datasets under the OOD evaluation.
> >
> > Some discussions on it might be beneficial.
> >
> > In general, I am happy about the responses and the revised version.
> >
> > I will raise the score to 6 accordingly.
> >
> > ----
> >
> > BTW, there is a typo on the caption of Table 2 (Line 338) "The shaded models are trained by othe rworks."

---

> > > ### Author Response · Authors · 2025-11-28
> > >
> > > Dear Reviewer UN8i,
> > >
> > > We sincerely thank you for your detailed feedback. Although, in this special case, the reviews will revert to the initial rebuttal process, we are still happy to see that you raised the score to 6. We have corrected the typo in the caption of Table 2 (Line 338) and updated the rebuttal revision. Regarding the performance on the OOD benchmark Drop, we believe that MaxFlow is more generalizable, which may explain why MaxFlow surpasses LCS on Drop. Further analysis will be more helpful for our upcoming improvements. Thanks again for your valuable comments to help us improve our paper.

---

### Official Review · Reviewer_PG2M · 2025-11-01

**Soundness:** 3
**Presentation:** 3
**Contribution:** 3
**Rating:** 6
**Confidence:** 4

**Summary:**

This paper studies structured reasoning for LLMs by (1) explicitly labeling reasoning steps with semantic tags, (2) extracting step-wise dependencies from attention weights, and (3) introducing two reinforcement learning rewards: a Max-Flow reasoning reward that encourages balanced step utility and a longest-common-subsequence (LCS)-based consistency reward. The method aims to inject structure into chain-of-thought processes and improve reasoning efficiency. Experiments on math and reasoning benchmarks show improvements over supervised and GRPO baselines, alongside qualitative analyses of attention structure.

**Strengths:**

1. The paper addresses an important problem: encouraging more structured and efficient reasoning in LLMs rather than unconstrained chain-of-thought generation.
2. The proposed Max-Flow reward is a creative attempt to use internal model signals (attention structures) to shape reasoning behavior, moving beyond pure outcome-based rewards.
3. Experiments are conducted on multiple reasoning benchmarks and across different model sizes, showing consistent improvements under constrained output budgets.

**Weaknesses:**

1. The approach heavily relies on attention maps reflecting causal reasoning order. This link is assumed rather than validated. No experiments demonstrate whether attention graphs truly correspond to semantic dependency structures, raising validity concerns.
2. Key components (structured tags, Max-Flow reward, LCS reward) are not fully isolated. It remains unclear which part contributes most. A more refined ablation (e.g., Max-Flow only vs LCS only vs tags only) would strengthen the causal claims.

**Questions:**

1. Table 1 max-length execution. How is the “max length” constraint applied in Table 1? hard decoding cap? truncation after generation?
2. Does attention truly correspond to reasoning structure?
3. The Max-Flow reward implicitly assumes that “better reasoning” corresponds to more balanced (not overly concentrated) attention flow across intermediate steps. Can the authors clarify the motivation for this assumption?

---

> ### Author Response · Authors · 2025-11-25
> **Response to Reviewer PG2M (1)**
>
> Thank you for your valuable feedback to help us improve our paper. We have revised our paper based on your feedback. We detail our response below and please kindly let us know if our response addresses your concerns.
>
> >  **Q1**:
> The approach heavily relies on attention maps reflecting causal reasoning order. This link is assumed rather than validated. No experiments demonstrate whether attention graphs truly correspond to semantic dependency structures, raising validity concerns. Does attention truly correspond to reasoning structure?
>
> **A1**: Yes, step-wise attenion achieves highly alignment with human reasoning annotations.
>
> We leverage the **Entailment Trees dataset** [1], which annotates gold-standard reasoning dependencies (premise → intermediate → conclusion) for ARC (AI2 Reasoning Challenge) science exam questions. After converting these examples into structured format, we represent ground-truth dependencies as binary adjacency matrices. To assess robustness, we evaluate on two reasoning scenarios: **Task 1 (no distractor)** contains only necessary reasoning steps, while **Task 2 (with distractor)** includes irrelevant information.
>
> Our validation employs a controlled comparison between two conditions: the **Structured** group preserves the logical reasoning order when feeding chains into the LLM, whereas the **Shuffled** group randomly reorders steps (fixing only question/answer positions) to disrupt the reasoning structure. For quantitative evaluation, we compute the **alignment score** as the proportion of gold dependencies where attention weight exceeds the average attention, and the **win rate** as the percentage of examples where the Structured group outperforms the Shuffled group.
>
> Across both tasks, the Structured group achieves significantly higher alignment with human annotations rather than spurious correlations.
>
> **Table R1** Comparison of Alignment between Stepwise Attention and Human Annotations
> | Task 1 (no distractor) | Avg Alignment | Win Rate | Task 2 (with distractor) | Avg Alignment | Win Rate |
> |------------------------|---------------|----------|--------------------------|---------------|----------|
> | Shuffled Group         | 28.48%        | 5.50%    | Shuffled Group           | 24.87%        | 4.71%    |
> | **Structured Group**   | **71.27%**    | **97.15%** | **Structured Group**   | **72.27%**    | **95.29%** |
>
> We have revised the paper and put more details in **Section 4.4**.
>
> [1] Explaining Answers with Entailment Trees, Dalvi et al., EMNLP 2021.

---

> ### Author Response · Authors · 2025-11-25
> **Response to Reviewer PG2M (2)**
>
> > **Q2**: Key components (structured tags, Max-Flow reward, LCS reward) are not fully isolated. It remains unclear which part contributes most. A more refined ablation (e.g., Max-Flow only vs LCS only vs tags only) would strengthen the causal claims.
>
> **A2**: In response to your suggestion and other reviewers' feedback, we conducted 9 controlled experiments covering both isolated and combined components. **We find that MaxFlow provides the largest contribution (+3.95% average gain), but structured tags are essential—without them, MaxFlow's effectiveness is severely limited (+0.73% only).**
>
> To systematically analyze each component's contribution, we designed two experimental groups:
>
> **Isolated Component Training (5 experiments):** (1) **+ Structured Tags** – Training with tags only; (2) **+ LCS** – Training LCS on free-form reasoning (extracting steps by `\n\n` and matching by similarity); (3) **+ MaxFlow** – Training MaxFlow on free-form reasoning (extracting steps by `\n\n`); (4) **+ Filtered Data GRPO** – Training GRPO only on filtered data; (5) **+ Dr.GRPO** – Training Dr.GRPO on free-form reasoning.
>
> **Combined Component Training (4 experiments):** (1) **Tags + GRPO**; (2) **Tags + LCS**; (3) **Tags + MaxFlow**; (4) **Tags + Step-Level MaxFlow**.
>
> **Table R2** Performance Improvements over Baseline
> | Method | 1K | 2K | 4K | 8K | Average |
> |--------|----:|----:|----:|----:|--------:|
> | + Structured Tags | +4.14 | +4.64 | +0.65 | +0.36 | +2.45 |
> | + LCS | +6.23 | +4.00 | +1.00 | +0.89 | +3.03 |
> | + MaxFlow | +3.39 | +1.25 | -0.64 | -1.09 | +0.73 |
> | + Filtered Data GRPO | +0.28 | +0.11 | +0.06 | -0.38 | +0.02 |
> | + Dr.GRPO | +4.22 | +4.97 | +1.31 | +1.37 | +2.97 |
> | Tags + GRPO | +5.18 | +6.25 | +2.35 | +0.28 | +3.52 |
> | Tags + Step-Level MaxFlow | +7.10 | +8.13 | +3.14 | +1.68 | +5.01 |
> | Tags + LCS (Ours) | **+10.79** | **+10.23** | +3.38 | -0.40 | +6.00 |
> | Tags + MaxFlow (Ours) | +8.45 | +8.17 | **+5.83** | **+3.12** | **+6.39** |
>
> Table R2 reveals clear component contributions and synergistic effects: Structured tags alone provide a +2.45% foundation. When combined with step-level rewards, **Tags + MaxFlow** achieves the best performance at **+6.39%** (an additional +3.94% gain), while **Tags + LCS** reaches **+6.00%** (+3.55% gain). Notably, isolated MaxFlow on free-form reasoning shows negative gains at longer contexts (4K: -0.64%, 8K: -1.09%), confirming that accurate step parsing via structured tags is prerequisite for effective step-level optimization.
>
> **Table R3** MaxFlow and LCS Performance Gains with/without Structured Tags
> | Response Length | MaxFlow on Structured | MaxFlow on Free-form | LCS on Structured | LCS on Free-form |
> |----------------|---------------------:|--------------------:|------------------:|------------------:|
> | 1K | **+4.31** | +3.39 | **+6.65** | +6.23 |
> | 2K | **+3.53** | +1.25 | **+5.59** | +4.00 |
> | 4K | **+5.18** | -0.64 | **+2.73** | +1.00 |
> | 8K | **+2.76** | -1.09 | -0.76 | **+0.89** |
> | **Average** | **+3.95** | +0.73 | **+3.55** | +3.03 |
>
>
> Table R3 further isolates the impact of structured tags: MaxFlow on structured reasoning achieves +3.95% average gain, while on free-form reasoning it gains only +0.73%. This performance gap demonstrates that structured tags enable proper step extraction and reward assignment, which is critical for MaxFlow's effectiveness.
> The detailed results are presented in **Appendix B** of the revised manuscript, clearly demonstrating each component's individual contribution and their synergistic effects.

---

> ### Author Response · Authors · 2025-11-25
> **Response to Reviewer PG2M (3)**
>
> > **Q3**: Table 1 max-length execution. How is the “max length” constraint applied in Table 1? hard decoding cap? truncation after generation?
>
> **A3**: The max-length constraint is enforced as a hard decoding cap during generation, which terminates when reaching the specified token limit or an end-of-sequence token, whichever comes first. We have revised the paper and put these details in **Section 5.1**.
>
> > **Q4**: The Max-Flow reward implicitly assumes that “better reasoning” corresponds to more balanced (not overly concentrated) attention flow across intermediate steps. Can the authors clarify the motivation for this assumption?
>
> **A4**: we propose the MaxFlow reward to encourage streamlined reasoning chains where all steps contribute meaningfully to the final answer. **(1) Unbalanced attention flow reveals redundant steps.** When attention is excessively concentrated on a few steps, it indicates that other steps contribute minimally to the final answer and are likely redundant or unnecessary. **(2) Balanced attention reflects essential dependencies.** In an ideal step-by-step reasoning, each step strongly depends on the previous one, making contributions approximately equal. Therefore, balanced (not overly concentrated) attention flow indicates a robust reasoning chain where all steps are necessary and non-redundant.
> We have revised the paper and provided additional details on the motivation in a newly added **Section 3**.

---

### Author Response · Authors · 2025-11-25
**Summary**

Dear Chairs and Reviewers,

We would like to thank the reviewers for their careful and constructive comments. We also thank the reviewers for acknowledging that our work presents interesting ideas (UN8i, 7R1m), demonstrates strong experimental results (PG2M, sQTy, 7R1m), and provides a well-motivated approach to reasoning optimization (sQTy).

The paper has been revised in accordance with the reviewers' comments and suggestions. Updates and changes are marked by green color in the revised version. Here are a few important points we would like to highlight:

**1. Motivation: Formulating Reasoning as Graph Optimization**

Free-form reasoning often suffers from redundant reasoning fragments intertwined with valid sentence, while structured reasoning enhances interpretability and enables more principled optimization.
To do so, we reformulate reasoning as a graph optimization problem where reasoning flows from question → steps → answer. To enable this graph-based optimization, we must first construct reliable reasoning graphs, then optimize them at different granularities. This perspective naturally motivates our three-component design:

- **(1) Structured Tags for Graph Construction:** To construct reasoning graphs, we need to reliably parse individual reasoning steps. Since small LLMs cannot consistently follow free-form training prompts, we introduce lightweight structured tags to provide stable step boundaries.

- **(2) MaxFlow  for Single-Graph Optimization:**  From a single-graph optimization perspective, we observe that reasoning steps with low flow from question to answer indicate redundant, repetitive, or irrelevant steps. Therefore, we propose the MaxFlow reward to encourage streamlined reasoning chains where all steps contribute meaningfully to the final answer.

- **(3) LCS reward for Multi-Graph Optimization:** From a multi-graph comparison perspective (multiple reasoning paths for the same question), we hypothesize that better reasoning graphs should (1) contain a higher proportion of optimal sub-paths shared across correct solutions, and (2) express each step more concisely. This motivates our LCS reward, which performs cross-path comparison to identify and reinforce common high-quality reasoning patterns.

**2. Structured Tag is important for MaxFlow**

In response to reviewers (PG2M, UN8i, 7R1m), we conducted detailed ablation studies. The key finding are as follows:

(1) **MaxFlow provides the largest performance gain (+3.95% average) with structured tags.** Structured tags are essential.  Without them, MaxFlow's effectiveness is severely limited (+0.73% only) due to inaccurate step parsing;

(2) Tags + MaxFlow consistently outperforms Tags + GRPO across all context lengths (1K-8K);

(3) Our method outperforms Dr.GRPO (+2.97%), which elegantly addresses length bias;

(4) Both LCS and MaxFlow show more stable performance across temperatures (0.1-1.0) compared to baseline methods. These experiments demonstrate the necessity of our integrated approach and clarify each component's contribution.

Should you need further information, please let us know. We look forward to hearing from you soon.

Yours sincerely,

Authors

---

### Meta-Review · Area_Chair_2uku · 2026-01-07

**Summary:**

All reviewers provided positive views about the paper, acknowledging that it addresses an important problem, introduces a creative strategy to influence the reasoning behaviour, and reports on comprehensive experimentation. Some reservations were expressed during the review process.

C1. The approach relies on attention maps reflecting causal reasoning order and there is no validation of this assumption. It is not clear that the attention graphs correspond to semantic dependencies.

C2. The study does not isolate key components of the proposed approach in a fine-grained ablation study.

C3. The two proposed variants do not exhibit consistent clear advantages over GRPO. There is no discussion explaining why this performance variability occurs.

C4. The stability and efficiency study is only conducted for one methodological variant.

C5. Important implementation details are not fully specified, making the overall approach difficult to evaluate and replicate.

**Reviewer Concerns:**

The response to the reviewers was comprehensive and detailed, with substantial revisions and additions to the paper. Three of the reviewers expressed satisfaction with the responses to their concerns. One of the most common criticisms (C2 – insufficient ablation study) was addressed by the addition of a much more extensive analysis. C1 was addressed by an experiment involving shuffled reasoning. C4 was addressed by an additional study. The authors provided the implementation details requested in C5 and clarified that the performance variability occurred in the OOD setting, where one would not expect uniform outperformance.

**Reviewer Scores:**

Reviewer PG2M. POSSIBLE BUT UNLIKELY CHANGE OF SCORE. The original score was 6, already expressing a positive view of the paper. While the authors provided a much more extensive ablation, addressing one of the two main criticisms, it is not clear if the reviewer would have been sufficiently convinced by the evidence in support of the relationship between causal reasoning and attention maps, particularly as the analysis was limited to a single dataset.

Reviewer sQTy. INDICATED AN INCREASE OF SCORE (from 4 to 6).

Reviewer UN8i. INDICATED NO CHANGE IN SCORE.

Reviewer 7R1m. INDICATED NO CHANGE IN SCORE.

---

### Decision · Program_Chairs · 2026-01-26

Accept (Poster)